# Towards Better Out-of-Distribution Generalization of Neural Algorithmic Reasoning Tasks

**Sadegh Mahdavi**                                                          *smahdavi@ece.ubc.ca*
*University of British Columbia*

**Kevin Swersky**                                                            *kswersky@google.com*
*Google Research, Brain Team*

**Thomas Kipf**                                                                *tkipf@google.com*
*Google Research, Brain Team*

**Milad Hashemi**                                                            *miladh@google.com*
*Google Research, Brain Team*

**Christos Thrampoulidis**                                               *cthrampo@ece.ubc.ca*
*University of British Columbia*

**Renjie Liao**                                                                 *rjliao@ece.ubc.ca*
*University of British Columbia*

**Reviewed on OpenReview:** *https: // openreview. net/ forum? id=xkrtvHlp3P*

## Abstract

In this paper, we study the OOD generalization of neural algorithmic reasoning tasks, where the goal is to learn an algorithm (*e.g.*, sorting, breadth-first search, and depth-first search) from input-output pairs using deep neural networks. First, we argue that OOD generalization in this setting is significantly different than common OOD settings. For example, some phenomena in OOD generalization of image classifications such as *accuracy on the line* are not observed here, and techniques such as data augmentation methods do not help as assumptions underlying many augmentation techniques are often violated. Second, we analyze the main challenges (*e.g.*, input distribution shift, non-representative data generation, and uninformative validation metrics) of the current leading benchmark, *i.e.*, CLRS (Veličković et al., 2021), which contains 30 algorithmic reasoning tasks. We propose several solutions, including a simple-yet-effective fix to the input distribution shift and improved data generation. Finally, we propose an attention-based 2WL-graph neural network (GNN) processor which complements message-passing GNNs so their combination outperforms the state-of-the-art model by a 3% margin averaged over all algorithms. Our code is available at: `https://github.com/smahdavi4/clrs`.

## 1 Introduction

Algorithms have been a vital part of today's computing systems, solving tasks ranging from scheduling to finding the shortest path in real-world maps. Algorithms often come with benefits like performance guarantees, provable correctness, and generalization to various instances. Designing algorithms to solve tasks, however, can be quite hard in many situations. For instance, given any task, a computer scientist has to either follow some existing algorithm paradigms (*e.g.*, divide and conquer) or come up with some new algorithm, which typically requires significant hand-design efforts. Moreover, some problems lie in the NP-hard class of which the solutions require an exponential amount of time for execution, and designing efficient approximate

algorithms for such problems often requires several years of research. Finally, executing an algorithm often requires preprocessing data into a predefined format before feeding the data into a computer program. On the other hand, deep learning has been successful in solving tasks of many domains without much task-specific adaptation. For instance, a Transformer (Vaswani et al., 2017) could solve image classification, machine translation, and language modeling with negligible changes to its architecture. Moreover, deep learning models are well known for their ability in solving tasks in an end-to-end fashion, thus removing the necessity of data preprocessing and feature design. Neural networks, however, are not resilient to distribution shifts, and their symbolic manipulation and reasoning capabilities are limited (Zhang et al., 2021).

Therefore, it is natural to ask whether one can leverage deep neural networks to mimic algorithms, *i.e.*, neural networks execute the same computation as given algorithms. The hope is to combine the strength of both worlds, *e.g.*, 1) performance guarantees, generalization to various instances, and transferability across tasks from algorithms, and 2) end-to-end training, universal approximation, and fast parallel computation from neural networks. Recently, researchers have been working on this problem and the area is dubbed as *Neural Algorithmic Reasoning* (Veličković & Blundell, 2021). Here, the goal is not only to enhance the symbolic manipulation and reasoning capabilities of neural networks but also to leverage them to find more efficient and faster algorithms.

Existing works have shown that this problem is challenging, especially when the model is evaluated on test samples that are drawn from a different distribution than train samples (*i.e.*, it is hard to generalize to out-of-distribution data) Veličković et al. (2021). For example, while deep neural networks could be trained to perform sorting on sequences of a given, fixed length, when faced with longer sequences, their performance drops substantially (Varis & Bojar, 2021). In this work, we analyze the out-of-distribution (OOD) generalization of neural networks on algorithmic reasoning tasks. We first investigate several challenges with OOD generalization in our problem context and propose solutions to some of the issues that arise. In particular, our contributions are as follows:

- In Section 3, based on the CLRS benchmark, we show how OOD settings of neural algorithmic reasoning tasks are different from common ones (Section 3.1) and identify several key challenges: input distribution shift (Section 3.2), unrepresentative dataset generation in both train and test (Section 3.3), model selection difficulties due to near-perfect in-distribution validation metrics (Section 3.4), and a systematic investigation of OOD generalization under controlled distribution shift of graph sizes and node degrees (Section 3.5).

- In Section 4, we propose solutions to some of these challenges, including 1) enlarging the in-distribution training dataset to push the limits of OOD performance (Section 4.1), 2) a simple-yet-effective fix to input distribution shift (Section 4.2), 3) an attention-based 2-WL graph neural network (GNN) (Morris et al., 2019) processor that significantly outperforms the state-of-the-art when combined with message passing GNNs (Sections 4.3 and 4.4 ).

## 2 Background

We follow the setup of the CLRS benchmark (Veličković et al., 2021). All algorithmic tasks are translated and represented as graphs (possibly empty or fully-connected) with task-specific node/edge/graph feature inputs and node/edge/graph-level target outputs. Each task also contains the ground truth values of all variables presented in intermediate steps of an algorithm, which are called *hints* and could be used as additional supervision to train the model. The aim of hints is to help mimic individual steps of the underlying algorithm and finally predict the target output.

Each instance of a task contains a graph $G = (\mathcal{V}, \mathcal{E})$ with a set of $n$ nodes $\mathcal{V} = \{v_1, v_2, \ldots, v_n\}$ and $m$ edges $\mathcal{E} = \{e_i = (v_{i_1}, v_{i_2}) : v_{i_1}, v_{i_2} \in \mathcal{V}, 1 \leq i \leq m\}$ with an adjacency matrix $A \in \mathbb{R}^{n \times n}$, edge feature tensor $X_{\mathcal{E}} \in \mathbb{R}^{n \times n \times d_{\mathcal{E}}}$, and node feature matrix $X_{\mathcal{V}} \in \mathbb{R}^{n \times d_{\mathcal{V}}}$. We eschew an explicit graph feature vector as it could be represented as a concatenation of features to the node feature matrix. For a node (column) vector $z_v$, we use capital from $Z$ to indicate a vertically stacked matrix (one row per node) of all node vectors across the graph. Moreover, we denote column-wise concatenation of two matrices/row-vectors $E$ and $F$ as $[\ E, \quad F\ ]$.

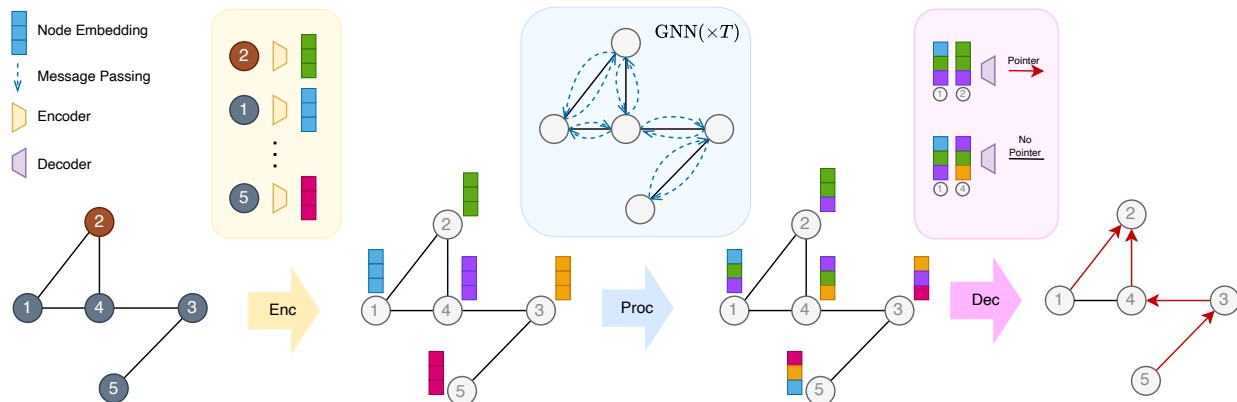

Figure 1: Encode-process-decode visualization for the BFS task. The input is a graph with starting node index of two. The node representations are encoded into a latent space. Then $T$ consecutive steps of processing are applied to the latent dimensions using a GNN. Finally, the output pointer for each node is decided by pairwise comparisons of node representations.

All models we consider are GNNs with an encode-process-decode framework (Battaglia et al., 2018). For an algorithm with $T$ iterative steps, we consider a recurrent GNN whole message passing steps correspond to algorithms steps. At each step $t$ ($1 \leq t \leq T$) of processing, all input features are embedded into embedding vectors $z_{\mathrm{inp}_v} \in \mathbb{R}^{d_h}$ by their corresponding encoders, for each individual node $v \in \mathcal{V}$. The processor then takes the embedding vectors and the hidden states at step $t$ and outputs the hidden representations of step $t+1$. Formally, a (simplified) formula for the processor is as follows:

$$H^{(t+1)} = \mathrm{GNN}([\ H^{(t)}, \quad Z_{\mathrm{inp}}\ ], A), \tag{1}$$

where $\mathrm{GNN}(X, A)$ represents the one-step computation process of a GNN processor with input node features $X$ and adjacency matrix $A$. Finally, outputs are decoded using a simple MLP per node/edge, with shared parameters. The loss value is evaluated based on the outputs of decoders at iteration $T$.

To better illustrate this framework, we explain how Breadth-first Search (BFS) is formulated in this framework. Given an input undirected graph and a starting node, the task is to predict for each node the pointer to its predecessor when applying BFS on this graph. Namely, we have two-dimensional node features $X_{\mathcal{V}} \in \mathbb{R}^{n \times 2}$ where the first dimension determines the index of a node and the second dimension determines whether the node is the starting node. We also have the adjacency matrix $A$ of the graph. The encoder is a linear layer with a weight matrix $W_{\mathrm{enc}} \in \mathbb{R}^{2 \times d_h}$ where $d_h$ is the latent embedding size. Then, for the processing step, $T$ recurrent steps of GNN are applied to the latent space with adjacency matrix $A$ to get the final latent node representations $H^{(T)} \in \mathbb{R}^{n \times d_h}$. Finally, for each pair of nodes $i, j$ ($1 \leq i, j \leq n$), a score $\alpha_{i,j}$ determines how likely $i$ points to $j$ as the predecessor and $\hat{y}_i = \arg\max_j \alpha_{i,j}$ is the predicted output pointer of node $i$. To compute $\alpha_i$, we can apply an MLP to $[h_i^{(t)}, h_j^{(t)}]$ followed by a (row-wise) Softmax non-linearity. The loss for this task is a cross-entropy loss between predicted row scores $\alpha_{i,:}$ and the true (one-hot row-wise) pointer scores for each node $i$. This process is illustrated in Figure 1. For additional examples, see Appendix A.3.

As the processor is considered the central part of this architecture, here we explain the three processor baselines used by the benchmark:

- **MPNN** (Gilmer et al., 2017): A message passing neural network with a max-aggregation. On the benchmark, this processor assumes a fully connected graph regardless of whether the downstream task contains any graph or not. We regard this fully-connected MPNN processor as **MPNN-FC**. We also consider the original MPNN which operates on given graphs (if graphs are unavailable for a task, a fully connected graph is used), and call it **MPNN-G**.

Table 1: Test scores of three processors (MPNN-FC, PGN, GAT) on the CLRS benchmark proposed by Veličković et al. (2021) on all 30 algorithms averaged over three random seeds. The performances with hints are lower on the majority of algorithms, for all processors. The mean is taken over all 30 algorithms, the wins are taken within each processor. For each algorithm, a win is declared when the score for that algorithm is higher than the alternative (*i.e.*, with vs. without hints).

| Method | MPNN-FC | | PGN | | GAT | |
|---|---|---|---|---|---|---|
| Use Hints | Yes | No | Yes | No | Yes | No |
| Mean | 51.02 | **60.04** | 52.31 | **57.02** | 44.69 | **49.53** |
| Wins | 8/30 | **22/30** | 10/30 | **20/30** | 14/30 | **16/30** |

- **PGN** (Veličković et al., 2020a): Similar to MPNN, but PGN operates on a graph derived from both the input and hints. When no hints are available, and the task does not contain any graph, it assumes an empty graph and hence behaves the same as DeepSets (Zaheer et al., 2017).

- **GAT** (Veličković et al., 2018): Similar to PGN, but GAT replaces the max-aggregation-based message passing in MPNN with the attention based one.

To fairly benchmark these processors, the encoder and the decoder are controlled to be the same.

## 3   Current Challenges of OOD generalization of Neural Algorithmic Reasoning Tasks

In the CLRS, Veličković et al. (2021) measures both the in-distribution (ID) and the out-of-distribution (OOD) generalization of the aforementioned processors over 30 classical algorithms. For example, they generate validation graphs from the same distribution as training but generate testing graphs from different distributions, *e.g.*, larger sizes in terms of the number of nodes. Table 1 contains the OOD test accuracies of the algorithms when using hints (i.e. intermediate steps). We also include the performance of the different models when disabling hints. As the table shows, in the current form of benchmark, hints do not bring any performance gain on the final output for the majority of algorithms. Moreover, to learn algorithms beyond those in the CLRS benchmark, hints might be unavailable or infeasible to collect at all (e.g., NP-hard problems). Based on these considerations, we deactivate hints for the rest of the paper and instead focus on end-to-end training with only the supervision of the final output. In the following, we highlight the fundamental challenges in the OOD generalization of neural algorithmic reasoning in the CLRS.

### 3.1   Difference with Common OOD Settings

Several recent works have studied the problem of OOD generalization in both image and graph domains (Arjovsky et al., 2019; Anil et al., 2022; Ovadia et al., 2019; Chen et al., 2022). However, the nature of OOD generalization in neural algorithmic reasoning is fundamentally different from these works. To illustrate an example, consider the phenomena of *accuracy on the line* (Miller et al., 2021) for image classifications where some works have observed the OOD performance has a linear relationship with (and often is close to) ID performance in some image classification benchmarks. However, in our setting, we often observe perfect ID generalization with poor (sometimes even less than random guesses) OOD accuracy, and OOD generalization might keep improving after the ID validation performance completely saturates (see Bridges task in Figure 2a). Moreover, popular data augmentation technique as an effective remedy for OOD generalization (Gulrajani & Lopez-Paz, 2021; Wenzel et al., 2022), are generally inapplicable to our setting. For instance, Mixup (Zhang et al., 2018) assumes convex combinations of inputs have the same convex combination of outputs, but for many algorithms, a convex combination of inputs from two tasks creates a new task with output significantly different from the convex combination of outputs from two tasks (*e.g.*, sorting the average of two sequences creates a whole new task, and the output is not a linear function of outputs of two tasks), thus violating the assumptions of Mixup. Even graph-specific data augmentation methods are generally inapplicable. For instance, DropEdge (Rong et al., 2020) assumes by randomly dropping an edge in the graph, the target

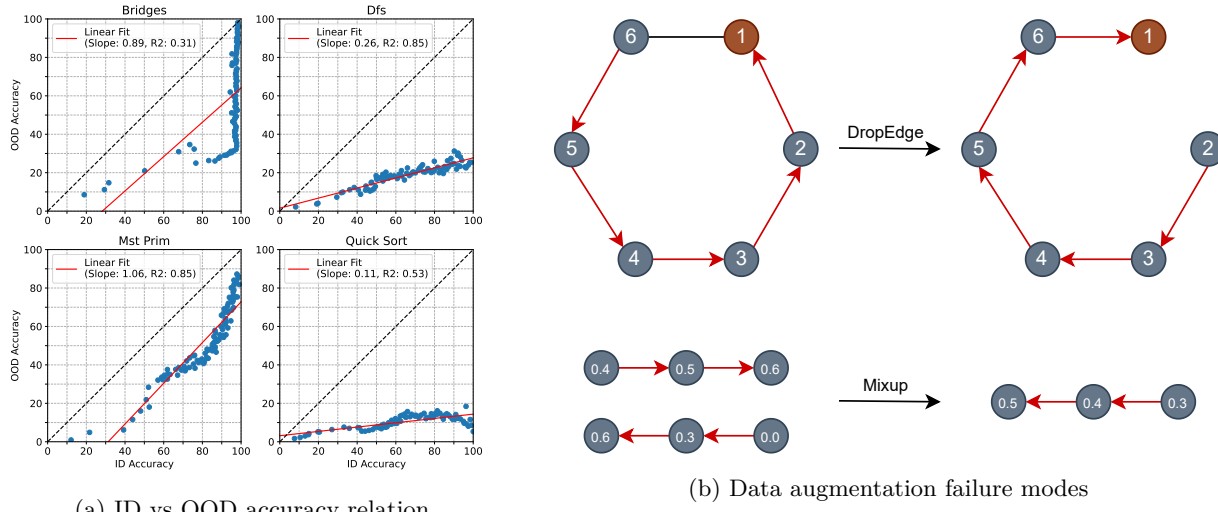

(a) ID vs OOD accuracy relation

(b) Data augmentation failure modes

Figure 2: (a) In-distribution (ID) accuracy (x-axis) vs OOD accuracy (y-axis) during different optimization steps for four different algorithms; in contrast to previous literature (Miller et al., 2021), either the linear correlation is non-existent, or the slope of the line is far from being one. (b) Failure of two widely-used data augmentation methods when applied to neural algorithmic reasoning tasks. Top: dropping a single edge ((Rong et al., 2020)) on DFS (starting from node 1) causes all of the output pointers in the graph (*i.e.* red arrows) to reverse direction. Bottom: using Mixup (Zhang et al., 2018) for sorting a convex node-wise combination of two sequences leads to a new sequence to sort, rather than the node-wise convex combination of labels of nodes of each sequence.

output stays the same. However, for many graph algorithms, altering any edge could change the output of the underlying task significantly. For example, performing DFS on a ring graph is an extreme failure example, since removing a single edge changes the outputs (*e.g.*, pointers of nodes to be visited next) of all nodes in the graph. See Figure 2b for an illustration of the failure modes of such examples.

A recent line of work has particularly investigated OOD generalization in GNN settings. However, the assumptions they make about generalization across environments (in our case graph sizes) are invalid to the neural algorithmic reasoning. Yehudai et al. (2021) studied such generalization from a local $k$-hop neighborhood perspective (where $k$ is the number of GNN layers) and connected failures in generalization to distribution shifts in the such local neighborhood. For algorithmic tasks, the numbers of reasoning steps are so large (sometimes larger than the number of nodes) that such a local neighborhood often contains the whole graph and the analysis would be vacuous. Some prior works have also analyzed the OOD generalization through the lens of graphons (Zhou et al., 2022; Bevilacqua et al., 2021; Maskey et al., 2022) where graphs along with labels are drawn from some grahpon models. In these settings, the implicit assumption is that if we restrict the original graph to a subgraph induced by a subset of its nodes, the labels within the subgraph stay the same. However, this favorable property does not hold in neural algorithmic reasoning tasks. For instance, consider the DFS example in Figure 2b. By removing node "2" and restricting the graph to the rest of the nodes, all the pointers turn into a clockwise order; hence the outputs change when restricting a graph to a smaller subset of its nodes. In algorithmic reasoning tasks, the relation between ID and OOD generalization is much more intricate than the above assumptions; the only connecting point between the target label of each task across different graph sizes is they are generated using the same algorithm. This makes even formulating the OOD generalization and the desired invariance properties in such tasks nontrivial. Although some prior works have discovered some desired invariance properties of a model for stronger OOD generalization of some algorithmic tasks, such as homogeneity of the neural network (Tang et al., 2020), finding sufficient conditions remains an open question.

### 3.2 Distribution Shift in Input Representations

Beyond improving models, finding suitable data representations to facilitate OOD generalization is also crucial. For instance, all algorithms in the CLRS benchmark receive the node indices as a part of their input. For several tasks, this node index serves as a unique flag (*e.g.*, Convex Hull, Bridges), for others, the ordering plays an important role in the algorithm behavior (*e.g.*, DFS algorithm starts with a node with the lowest index, then, at each step, an unexplored neighbor with the lowest index gets expanded). As a result, the model must be able to compare node indices to be able to execute the algorithm.

The current node index implemented in the CLRS (Veličković et al., 2021) appends a scalar $\text{pos}_{v_i} = (i-1)/n$ to the feature vector of each node $v_i$ with index $i, (1 \leq i \leq n)$. This kind of representation makes it harder for a processor to generalize when such ordering is important in a task. For instance, despite the model being able to reach perfect ID accuracy to execute DFS on 16-node graphs; it fails to do so when we feed the induced subgraph of the first quarter nodes of a 64-node graph. This is while the graphs are still in-distribution, and the only difference is that the indices have shifted from 16 equispaced indices in the range $[0, 1]$ to 16 equispaced indices in the range $[0, 0.25]$. In Section 4.2 we explore two different possible mitigations to such a distribution shift for node indices.

### 3.3 Unrepresentative Dataset Generation

The graphs in the CLRS benchmark are sampled using Erdős–Rényi (ER) model with a fixed probability $p$ (where $p = 0.5$ in most cases) and node/edge/graph-level scalar input features are generated uniformly at random (mostly from $[0, 1]$). In this strategy, sampling is uniform and all possible graphs (of the same size) have an equal probability of being present in the dataset. However, it takes many samples for this random graph model to sample diverse graphs (*e.g.*, node-degree and the number of edges in ERs concentrate around $(n-1)p$ and $n(n-1)p/2$ respectively). This raises two issues, especially in the low-data regime (*i.e.*, fewer training data).

First, the training data might not be enough to uniquely identify the underlying algorithm (*e.g.*, BFS and DFS have the same output pointers on the class of tree graphs – all nodes point to their parents towards the root) and the OOD generalization should not be expected in this case.

Second, and perhaps more importantly, the test dataset may turn into a poor performance measure as it is not diverse enough and biased. This bias may allow cheating solutions to gain higher scores, especially when extrapolating (Angelini & Ricci-Tersenghi, 2022). For instance, two of the most challenging algorithmic tasks in the CLRS are Articulation Points and Bridges. In these tasks, the goal is to identify the nodes/edges whose removal increases the number of connected components of the underlying graph. Here, if a graph is too sparse, trivial bridges/articulation points appear in the graph. On the other hand, a dense graph leads to no such nodes/edges. In Section 4.4, we report a 98.69% OOD accuracy on the Bridges task using a hybrid combination of 2WL and MPNN-G processor. At first sight, the Bridges algorithm seems to be completely solved and the model can reliably find the bridges in a graph. However, the test dataset is not rich enough for testing the model. The bridges in the dataset are mostly chain subgraphs branching out of a connected component (see Section 5 for more discussion).

### 3.4 Model Selection is Hard with Nearly Perfect ID Performances

Generally, model selection in OOD settings is not as straightforward as in the ID setting. This is because the validation data is not identically distributed as the test data. This problem is accentuated even more in algorithmic reasoning tasks. When training a model on algorithmic tasks of the CLRS benchmark, it usually fits the ID validation data almost perfectly in terms of prediction accuracy. This leaves a harder model selection question than common OOD settings, *i.e.*, "How to pick the best model among all these perfect models?". These perfect-ID models behave very differently on OOD data. For example, despite all the architectures achieving 100% ID accuracy on the Articulation Points task (Table 9 in the Appendix), the OOD accuracies widely differ across different architectures, ranging from 69% to 88%. This issue exists even when comparing models from the same architecture but in different optimization steps. For example, for some tasks, the OOD accuracy keeps improving even after perfectly fitting the ID data. An extreme case can

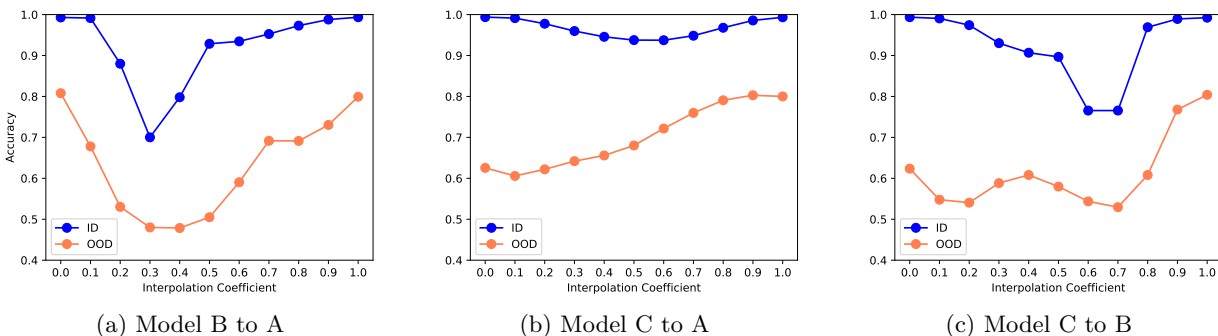

(a) Model B to A          (b) Model C to A          (c) Model C to B

Figure 3: Model interpolation on the Matrix Chain Order task. We train three models (named as A, B, C) and pairwise interpolate their weights to obtain ID and OOD accuracies. Each plot shows the interpolation of a pair (*e.g.*, when the coefficient is set to 0 and 1 in (a), we use model B and A respectively; when it is in the middle, we linearly interpolate weights from B and A). While models achieve similar ID accuracy, their interpolations show very different OOD performances.

be seen in Figure 2a for the Bridges algorithm. Here, the OOD accuracy keeps improving from around 30% to above 90% after an almost perfect ID performance is achieved.

Furthermore, OOD performances in some algorithms have a high variance across models, and sometimes even negatively correlate with ID performances. Inspired by recent works on mode-connectivity and model weight interpolation (Wortsman et al., 2022; Entezari et al., 2022), we interpolate between the weights of models to better understand the loss landscape of such tasks. We pick the Matrix Chain Order task which shows a high variance even when the initialization point of the model is fixed. Here, despite all models attaining 99.5% ID accuracy, some of them have an OOD accuracy of around 60%, and some have an OOD accuracy of around 80%. Figure 3 shows pairwise interpolation of three models trained on this task. According to the figure, there are barriers when connecting two sets of weights. Quite counter-intuitively, the ID barrier between two models with a larger OOD gap is the smallest (Figure 3b). Moreover, there are two barriers for the ID performance, and one barrier for the OOD in Figure 3c. Finally, when interpolating from model C to model A, the ID and OOD do not show any positive correlation and even negative correlation in the first half of the plot (3b). These all support the fact that the ID validation performance is not a suitable measure for model selection, and ID and OOD data clearly have different loss landscapes. See Figure 5 in the Appendix for additional interpolation plots.

### 3.5 Disentangling Different OOD Generalization Behaviors

The CLRS benchmark (Veličković et al., 2021) generates the underlying graphs using an Erdős–Rényi model with a fixed probability $p$. This means that the node degree distribution increases as the graph size increases. While prior works have worked on proving OOD generalization guarantees when the local structure of graphs stays the same (Yehudai et al., 2021), neural algorithmic reasoning is different from standard graph benchmarks. Here, when the graph size increases, higher-depth reasoning is needed. As a result, a fixed-depth local structure of the graph is no longer sufficient for a node to decide the label. To disentangle these two types of generalization, we create three dataset configurations of the CLRS benchmark to better understand such generalization bottlenecks.

We consider only graph algorithms since it is not possible to control the message-passing locality of sequence/set algorithms. There are 11 distinct graph algorithms in the benchmark. We remove the Articulation Points and Bridges algorithms since they require complicated sampling/evaluation strategies for preventing cheating solutions from succeeding. We create the following datasets:

Table 2: Test scores of MPNN-G processor on controlled datasets of 9 graph algorithms, *i.e.*, L-CLRS-Len, L-CLRS-Deg, and L-CLRS-Len-Deg.

| Accuracy Metric | Node Level | | | Graph Level | | |
| Method | CLRS-Len-Deg | CLRS-Len | CLRS-Deg | CLRS-Len-Deg | CLRS-Len | CLRS-Deg |
| --- | --- | --- | --- | --- | --- | --- |
| Bellman Ford | 99.48± 0.07 | 99.57± 0.06 | 99.40± 0.09 | 84.93± 2.24 | 87.53± 1.69 | 82.97± 2.11 |
| Bfs | 100.00± 0.00 | 99.92± 0.04 | 100.00± 0.00 | 100.00± 0.00 | 97.77± 1.23 | 100.00± 0.00 |
| Dag Shortest Paths | 99.84± 0.04 | 99.94± 0.00 | 99.81± 0.02 | 95.10± 1.18 | 98.17± 0.12 | 94.37± 0.66 |
| Dfs | 74.63± 3.91 | 81.76± 3.66 | 98.91± 0.23 | 0.13± 0.12 | 3.67± 1.68 | 81.57± 2.82 |
| Floyd Warshall | 63.84± 1.07 | 64.51± 0.34 | 51.98± 2.88 | 0.00± 0.00 | 0.00± 0.00 | 0.00± 0.00 |
| Mst Kruskal | 49.80± 1.54 | 61.66± 1.56 | 51.69± 1.53 | 0.00± 0.00 | 0.00± 0.00 | 0.00± 0.00 |
| Mst Prim | 87.02± 0.90 | 95.33± 0.42 | 85.72± 1.56 | 6.67± 1.87 | 53.20± 3.03 | 4.47± 2.15 |
| Strongly Connected Components | 94.45± 0.39 | 94.18± 0.04 | 96.19± 0.09 | 65.57± 0.75 | 64.40± 0.96 | 75.63± 1.76 |
| Topological Sort | 94.30± 0.27 | 93.76± 0.01 | 98.56± 0.09 | 54.12± 0.78 | 52.02± 0.25 | 80.97± 0.67 |
| Mean | 84.82 | 87.85 | 86.92 | 45.17 | 50.75 | 57.77 |

- L-CLRS-Len: We control the degree distribution of nodes between training and testing to be the same via sampling random $K$-regular graphs. Only the size (*i.e.*, number of nodes) of testing graphs is larger than those in training. This is an ideal setting for studying size generalization.

- L-CLRS-Deg: We control the number of nodes between training and testing to be the same. Only the degree distribution of nodes changes, which is achieved by changing the $K$ value in $K$-regular graphs. This is ideal for studying how the degree distribution shift affects the generalization.

- L-CLRS-Len-Deg: We allow both the size and the degree distribution to be changed between training and testing, which is a setting closest to the original CLRS. It is useful to study both size and degree generalization. Here, in the test set, we not only increase the number of nodes but also increase the $K$ value of $K$-regular graphs.

The details of these datasets are summarized in Table 3. Note that the motivation behind choosing K-regular graphs is that degree shift can be completely controlled and no shift in the variance of degree. Moreover, dataset leakage is avoided as we generate more training graphs (*i.e.*, graphs in the test dataset have zero probability of appearing in the training set). In addition to the CLRS's own metric (*i.e.*, node-level metrics), we include a graph-level metric that is stricter than the node-level one. In particular, a graph is considered as correct if all of its nodes are classified correctly (in contrast to the node-level metric which measures the accuracy of the node classification). This way, we can better distinguish performances among models.

We observe that while some algorithms struggle more on size generalization (*e.g.*, DFS), others struggle on degree generalization (*e.g.*, MST algorithms, Floyd Warshall). This suggests that, even if we control all the distribution shifts caused by size generalization, distribution shift in degree is still challenging for some algorithms and vice versa. Moreover, the graph-level metric shows very poor performance in some tasks, despite a high node-level performance (*e.g.*, Floyd Warshall, and MST algorithms). This shows that in tasks with low graph-level accuracies, all graphs contain challenging parts. Finally, lower graph-level and node-level accuracies of CLRS-Len-Deg compared to the other two datasets confirms that when two distribution shifts are combined, the prediction problem becomes more challenging.

## 4 Improving OOD Generalization of Neural Algorithmic Reasoning

In this section, we introduce our solutions to some of the above challenges that improve the performance of end-to-end (*i.e.*, without hints) neural algorithmic reasoning. Since hints are disabled in our case, we make two changes to our model. First, algorithms with the same input/output (*e.g.*, Dijkstra and Bellman-Ford) only differ in the number of processing steps. Since we discard these intermediate steps, the number of steps becomes meaningless and we use a number of steps independent of the underlying algorithm. Moreover, the number of tasks shrinks from 30 to 24 tasks (*e.g.*, Dijkstra and Bellman-Ford become Shortest Path, Quick/Insertion/Merge/Bubble/Heap Sort become Sort, *etc.*). In order to facilitate the comparison, we do

Table 3: Summary of dataset configurations used in this paper.

| Dataset | Lengths | | Dataset Size | | Dataset Generation |
|---|---|---|---|---|---|
| | Train/Val | Test | Train | Val/Test | |
| CLRS(Veličković et al., 2021) | 16 | 64 | 1000 | 32 | Erdős–Rényi graphs with fixed $p$ |
| L-CLRS | 16 | 64 | 100 000 | 32 | Erdős–Rényi graphs with fixed $p$ |
| L-CLRS-Len | 16 | 32 | 100 000 | 1000 | K-Regular graphs with $K = 4$ |
| L-CLRS-Deg | 32 | 32 | 100 000 | 1000 | K-Regular graphs with $K = 4$ for train/val, and $K = 8$ for test |
| L-CLRS-Len-Deg | 16 | 32 | 100 000 | 1000 | K-Regular graphs with $K = 4$ for train/val, and $K = 8$ for test |

not change the names of tasks and keep their original names from the CLRS benchmark. Second, we use MPNN-G as the message-passing processor.

## 4.1 Increase Dataset Size and Representativeness

First, we propose to increase the dataset size. The purpose of this change is to truly measure the extrapolation properties of each model and avoid some in-distribution training issues like overfitting. We change the CLRS dataset configuration and increase the amount of ID training samples from 1000 to $100K$ (and keep the ID validation and OOD test sets as they are). This way, it is possible to observe a much more diverse set of ID graphs. We note this strategy as a remedy to the low-data regime and not a solution. This strategy especially becomes problematic and even more data-hungry as the size of training graphs grows, and sampling diverse graphs becomes harder since the graph statistics become even more concentrated (*e.g.*, as the size of the graph grows, the number of samples to generate a disconnected graph increases). While this strategy is not a strong solution to OOD generalization, it helps mitigate any side factors contributing to low benchmark numbers but not directly related to size generalization; hence, providing a better test bed for testing size generalization. Table 3 summarizes the different datasets we use in this paper and their comparisons.

## 4.2 Improving Representation Distribution Shift

The position index of each node is a vital part of the inputs, especially since several algorithms depend on the ordering of such positions. Here, we explore a few possibilities of generalization to larger sizes. A suitable representation must be naturally extendable to any size of a graph. For instance, a one-hot encoding of indices is not generalizable as it contains a fixed input length and the input dimension is not fixed across different sizes.

As baselines, we consider the default index encoding used in the CLRS benchmark (*i.e.*, a scalar feature for each node $i$ as $(i - 1)/n$ where $n$ is the number of nodes). We also add positional encoding used by Transformers (Vaswani et al., 2017) as a baseline. We propose the following strategy as a mitigation to the distribution shift of the scalar index:

**Random Scalar Index.** Although the Scalar Index representation is naturally extendable to any graph size and node orderings are preserved, the model might rely on spurious correlations between such numbers (*e.g.*, the distribution shift due to moving to larger sizes makes generalization harder). For instance, the model might rely on the fact that consecutive indices always have a difference of $1/n$ where $n$ is the training dataset size, hence struggling to generalize as the size of graph grows and the difference between two consecutive indices becomes smaller. To mitigate the distribution shift, we propose to randomly sample scalars from $[0, 1]$ and use its sorted sequence during training, and use the same Scalar Index for validation and testing as mentioned before. This way, the model observes a variety of position embeddings and learns to better preserve the ordering of positions. In our experiments, we find that using random scalar makes training harder (*e.g.*, convergence is slower). As a remedy, we use a mix of both random and deterministic scalar index in a mini-batch as a training strategy to ease the training and achieve a faster convergence (*i.e.*, in each mini-batch, with the probability of 0.5 for each graph, we replace the position index of the whole graph with a random scalar index).

Table 4: Performance of different positional encoding strategies on positions-order aware algorithms using MPNN-G processor on L-CLRS dataset.

| Method | Scalar | Random Scalar | Position Enc. |
|---|---|---|---|
| Dfs | 25.36± 0.17 | 31.71± 1.13 | 11.90± 0.89 |
| Find Maximum Subarray Kadane | 26.60± 5.95 | 47.85± 5.32 | 12.97± 3.96 |
| Dag Shortest Paths | 99.82± 0.06 | 99.79± 0.02 | 31.45± 2.69 |
| Naive String Matcher | 8.24± 2.33 | 7.67± 3.94 | 1.32± 0.35 |
| Matrix Chain Order | 80.69± 1.33 | 92.07± 0.11 | 30.96± 3.16 |
| Topological Sort | 82.62± 0.63 | 84.17± 1.67 | 33.28± 0.84 |
| Bfs | 100.00± 0.00 | 100.00± 0.00 | 49.40± 4.83 |
| Mean | 60.47 | **66.18** | 24.47 |

In Appendix A.4, we also propose a more invariant (to shift and scale) way of representation, however, due to the superiority of the Random Scalar Index, we choose Random Scalar for all remaining experiments.

### 4.3 An Attention-Based Processor Competitive with MPNN-G

Inspired by recent works on higher-order message passing beyond node-level message passing, we explore this approach as a processor (Bodnar et al., 2021b; Kim et al., 2022; Morris et al., 2019; Morris & Mutzel, 2019). We treat each edge of the input graph as a node, construct a new graph (*i.e.*, *line graph* in graph theory) and run a GNN on top of that. Namely, the set of nodes, edges, and features $\mathcal{V}', \mathcal{E}', X'$ will be as follows:

$$\mathcal{V}' = \{v'_i = (v_{i_1}, v_{i_2}) : v_{i_1}, v_{i_2} \in \mathcal{V}, 1 \le i \le m\}$$
$$\mathcal{E}' = \{e'_i = (e_{i_1}, e_{i_2}) : e_{i_1}, e_{i_2} \text{ share a common node }, 1 \le i \le m'\}$$
$$X' = \{x'_i = [\ x^{\mathsf{T}}_{v_{i_1}}, \quad x^{\mathsf{T}}_{v_{i_2}}, \quad x^{\mathsf{T}}_{e_i}\ ]^{\mathsf{T}} : v_{i_1}, v_{i_2} \in \mathcal{V}, e_i \in \mathcal{E}, 1 \le i \le m\}.$$

In other words, nodes of the new graph are the edges of the original graph, edges are between every two original edges that share a node, and node features are concatenations of each endpoint of each edge. We use a transformer with edge masks of $\mathcal{E}'$ as the GNN. Such a GNN architecture has a similar computation mechanism as a 2-Weisfeiler-Lehman (2WL) test, and hence we call this processor "2WL". We, however, note that while the processors are limited by the expressivity of K-WL test (Morris & Mutzel, 2019), our model as a whole does not have such a limitation since in our setting node features are equipped with a unique identifier (Fereydounian et al., 2022). The result of using this processor is present in Table 5. As the results suggest, this processor shows competitive results with MPNN-G, and outperforms MPNN-G in some algorithms, while under-perming MPNN-G in others. For instance, the 2WL processor shows a 20% higher performance on the String Matching problem, while the MPNN-G shows a 20% higher performance on the Articulation Points finding algorithm.

### 4.4 Hybrid Processor

As we saw above, different processors exhibit different generalization behaviors. Here, we show a hybrid processor brings performance gains over individual ones. To this end, we take MPNN-G and 2WL processors as they show competitive results, and are quite complementary in nature (*e.g.*, max aggregation vs. attention, edge-wise message passing vs. node-wise message-passing). We consider the following processor combination strategies:

- **Average**: At each individual step of processing, two processors process the hidden representation, and the representation of each node/edge is computed using an average of two processors. Namely,

$$h^{(\mathrm{P})}_{v_i} = \frac{h^{(\mathrm{P}_1)}_{v_i} + h^{(\mathrm{P}_2)}_{v_i}}{2} \quad \text{and} \quad h^{(\mathrm{P})}_{e_i} = \frac{h^{(\mathrm{P}_1)}_{e_i} + h^{(\mathrm{P}_2)}_{e_i}}{2},$$

where $\mathrm{P}_1, \mathrm{P}_2, \mathrm{P}$ are two processors and the hybrid hidden representations.

Table 5: Mean test scores of different processors averaged over all 24 distinct algorithms on the L-CLRS dataset (performances of individual algorithms are averaged over three random seeds). The two rows correspond to using the scalar index and the random scalar index for input representation respectively. The hybrid models are a combination of MPNN-G and 2WL with the same parameter budget as individual processors (Table 13 and 14 contain the complete test scores of individual algorithms).

| Method | MPNN-G | 2WL | Hybrid-Average | Hybrid-Sigmoid |
|---|---|---|---|---|
| Scalar Index | 69.23 | 67.96 | 70.92 | **71.98** |
| Random Scalar Index | 71.11 | 71.52 | **74.33** | 74.31 |

Table 6: Summary of sequentially applying the improvements proposed in this paper. Test scores are averaged over all 24 distinct algorithms.

| Method | Mean Test Score | Improvement (%) |
|---|---|---|
| Baseline | 60.93 | - |
| Increase ID Dataset Size | 69.23 | 8.30 |
| Improve Input Representation Shift | 71.11 | 1.88 |
| Add Hybrid Processor | **74.33** | 3.22 |

- **Sigmoid**: Similar to average, but a sigmoid is applied to the previous hidden dimension to assign a weighted combination of processor hidden representation values. That is,

$$h_{v_i}^{(\mathrm{P})} = \sigma\left(h_{v_i}^{(\mathrm{P})}W\right) h_{v_i}^{(\mathrm{P_1})} + \left(1 - \sigma\left(h_{v_i}^{(\mathrm{En})}W\right)\right) h_{v_i}^{(\mathrm{P_2})}$$

$$h_{e_i}^{(\mathrm{P})} = \sigma\left(h_{e_i}^{(\mathrm{P})}W\right) h_{e_i}^{(\mathrm{P_1})} + \left(1 - \sigma\left(h_{e_i}^{(\mathrm{En})}W\right)\right) h_{e_i}^{(\mathrm{P_2})},$$

where $W$ is a $d \times 1$ weight matrix.

This processor is trained from scratch. Table 5 shows the results of experiments with these processors. As the results suggest, a hybrid processor does bring an average accuracy gain on performance. Furthermore, to show that the boost in the results indeed originates from the complementariness of processors, we also report the results of a hybrid model of two processors of the same type (*e.g.*, hybrid of two separate 2WL processors) in Table 12 in the Appendix. Next, we combine the proposed random scalar index representation and the 2WL processor in Section 4.3 and report the average score on the OOD testing dataset in the bottom row of Table 5. The hybrid models already outperform individual processors, and by adding the proposed input representation change, all processors gain $2\% - 3\%$ improvement on the average accuracy. The hybrid processors achieve an average score of above 74% on the benchmark. A unified summary table of improvements proposed in our work is shown in Table 6.

## 5 Discussion on OOD Performances

So far, we have made a series of improvements to the models and training data and improved the OOD accuracy of several tasks. Indeed, we have even achieved OOD accuracy above 98% on some tasks. Here, we ask "Is a task solved given our model achieves a high score on the benchmark for that particular task?". As discussed in Section 3.3, the test dataset might be biased and all graphs in the test set might have a common property and the model might take advantage of that property to output solutions. Following this issue, we pick the Bridges task on which the hybrid of MPNN-G and 2WL achieves a high OOD performance as shown in Table 14. As we show in Figure 4a, the bridges are all chain subgraphs branching out of a connected component.

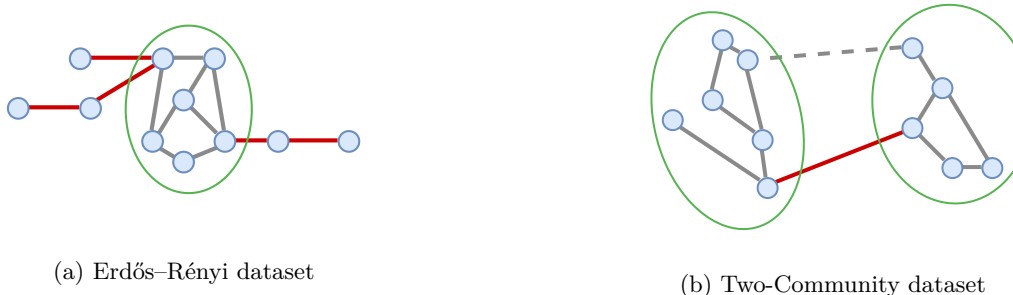

(a) Erdős–Rényi dataset

(b) Two-Community dataset

Figure 4: Illustration of two datasets generated for measuring the performance of a model on Bridges task. (a) The dataset generated by the CLRS benchmark. (b) We generate a two-community dataset for a stricter measurement of the Bridges task. The dashed edge is present in one graph and absent in the other graph of the pair. Red edges show the target bridges. Green ovals show connected components. A model with 98% OOD F1 score on the dataset (a) fails to detect the bridge between the two communities of the dataset (b).

To better evaluate models in this task, we create pairs of two-community graphs. Each pair of the community graph has two Erdős–Rényi generated and inter-connected subgraphs. Then, we connect the two communities using a randomly chosen edge. We also connect the two communities of the second graph in the pair using another random edge (see Figure 4b). This way, the first edge is a bridge in one graph and not a bridge in the second graph due to the existence of another edge connecting the communities. Ideally, a model capable of reasoning on bridges task must be able to identify the existence of the mentioned bridge in one graph, and the absence of the bridge in the second graph of the pair. Although there might be more bridges among the pairs of graphs (*i.e.*, inside individual communities), we do not consider them in our metric since most of them might be trivial. Therefore, a random guess gives 25% accuracy in such an evaluation. We evaluate the model on 1000 randomly generated pairs of graphs, we find out that in only seven pairs of graphs, the model detects the existence of the mentioned bridge in both graphs correctly. This 0.70% accuracy is much lower than the 25% chance accuracy. This suggests that despite the remarkable improvement on the benchmark (originally from 72% to 98%), there is still a lot of room for improvement, and OOD generalization in the Bridges task is not even close to being solved. Therefore, as the models get stronger, test datasets must also be refined to not only better evaluate the reasoning capabilities of a model, but also prevent heuristics from achieving high scores.

## 6    Related Works

In addition to the related works discussed in Section 3.1, there are some lines of research close to this area that we explain below.

**Neural Algorithm Execution.** A recent line of work has focused on the execution of algorithmic tasks using neural networks (Del'etang et al., 2022; Joshi et al., 2022; Abbe et al., 2022). Some works consider NP-Hard problems over graphs and propose problem-specific solutions (Karalias & Loukas, 2020; Bresson & Laurent, 2021), however, perfect extrapolation is infeasible for such problems (Yehuda et al., 2020). Some works consider executing algorithmic tasks, however, they are either limited to simple arithmetic tasks (Kaiser & Sutskever, 2016) or are general purpose but hard-to-train and struggle to even fit in-distribution data for complex tasks (Graves et al., 2014). Yan et al. (2020) proposed a general neural executor to mimic elementary subroutines of any algorithm, but requires low-level design and assembly of subroutine engines for each specific task. More recent works have proposed general-purpose GNN pipelines for executing (graph) algorithms (Veličković et al., 2020b;a). Several benchmarks have been proposed to measure the reasoning capabilities of neural networks (Zhang et al., 2022; 2021; Veličković et al., 2021). Perhaps the CLRS benchmark (Veličković et al., 2021) is the most comprehensive benchmark to date, as it contains 30 highly complex and challenging algorithmic reasoning tasks. In a concurrent work, Ibarz et al. (2022) proposed a series of improvements for improving OOD generalization when hints are used.

**Expressivity of GNNs.** Previous works have studied the expressiveness of GNNs from a WL-test point of view (Weisfeiler & Leman, 1968; Xu et al., 2019; Morris et al., 2019; 2021). These works show negative results by proving message passing GNNs are unable to distinguish two graphs if the WL test fails in distinguishing them. There exists a considerable body of literature for lifting such a limitation and improving the expressivity of GNNs by proposing novel architectures and providing analysis (Qian et al., 2022; Bevilacqua et al., 2022; Bodnar et al., 2021b;a; Maron et al., 2019). Some prior works have also addressed the limitations of GNNs by investigating their power in solving algorithmic tasks such as substructure counting in graphs (Bouritsas et al., 2022; Chen et al., 2020). All these limitations on the expressivity of GNNs, however, can be resolved by augmenting node features (Dasoulas et al., 2020; Fereydounian et al., 2022), at the cost of losing guaranteed equivariance. Since all tasks in our setting contain unique identifiers, none of the above limitations are applicable to our setting. This can also be verified by looking at validation scores. Here, the challenge is to extrapolate to larger graphs, rather than the possibility of learning to solve an in-distribution task.

**Reasoning Capabilities of Language Models.** With the success of Large Language Models (LLMs) in natural language understanding tasks, a series of recent studies have focused on extending the reasoning capability of LLMs even further. The core of these models is a Transformer architecture, which makes such tasks in nature very close to neural algorithmic reasoning tasks. Recent works improve the performance of LLMs using fine-tuning, chain-of-thought prompting (Kojima et al., 2022; Wei et al., 2022b; Anil et al., 2022), sampling multiple solutions (Lewkowycz et al., 2022; Wang et al., 2022), and scaling (Wei et al., 2022a). The most relevant idea to our setting is chain-of-thought prompting, which forces the model to do multi-step reasoning and is close in nature to the purpose of hints, but, in contrast to LLMs, hints do not bring any noticeable accuracy gain in neural algorithmic reasoning. Scaling the models might also be relevant, however, in our setting, the depth of the transformer must also increase with the size of the diameter of training graphs, which makes training large models on large graphs highly challenging.

## 7    Conclusion

In this work, we investigate the OOD generalization of neural algorithmic reasoning tasks from an end-to-end training perspective. We identify and analyze several challenges of OOD generalization under the current framework. Then, we propose solutions to some of these challenges by making the dataset larger, designing fixes to input representation distribution shift, and proposing an attention-based processor. These improvements, lead to a final 74.33% accuracy averaged over all algorithms. Finally, we shed light on the limitation of our model in executing one of the benchmark tasks and conclude that despite our model gains a high OOD test accuracy, there still exists much room for improvement. Our analysis opens up several interesting directions for future work. Designing better hints with guaranteed invariance properties can potentially facilitate the step-by-step execution of algorithms and help improve OOD generalization. Sampling training graphs from a diverse set of distributions could help boost the generalization. Improving the model selection strategy and finding an alternative to the validation set could help pick the best-performing models. Finally, it would be interesting to develop a meta-learning framework for inductive learning across graph sizes, in order for the model to learn the invariance properties of an algorithm across different sizes and generalize to larger graphs.

**Acknowledgments**

This work was funded, in part, by NSERC DG Grants (No. RGPIN-2022-04636 and No. RGPIN-2021-03677), Google Cloud credits, and Oracle Cloud credits. Resources used in preparing this research were provided, in part, by Google, Advanced Research Computing at the University of British Columbia, the Oracle for Research program, and the Digital Research Alliance of Canada (`alliancecan.ca`).

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

## A    Appendix

### A.1    Experiment Details

For Table 1 we take the (with-hint versions) numbers from Veličković et al. (2021) and run the no-hint version of the table with the same configuration except for disabling hints.

For the rest of the experiments, we use a batch size of 32 for message-passing processors and 16 for high-order processors, gradient clipping to a norm 1.0, a learning rate of 0.0001, 20 000 optimization steps, and cosine annealing scheduler. Moreover, as the number of processing steps is meaningless in the no-hint version, we set this number to 32 (*i.e.* 32 recurrent steps of message passing). Unless otherwise specified, experiments are run using the MPNN-G processor on the L-CLRS dataset. Finally, as the ID accuracy and OOD accuracy are not highly correlated, we do not do any early-stopping and among the models with the highest validation accuracy, we keep the last model.

The number of distinct algorithms shrinks from 30 to 24 when doing such changes. In particular, the following algorithms merge together:

- Sorting: All variants of sorting. That is, Quick sort, Insertion sort, Bubble sort, Heap sort are merged together. We Keep the "Quick sort" name to represent sorting and refrain from creating new names.

- Single source shortest path: Dijkstra and Bellman-Ford are merged together. We keep the "Bellman Ford" name to represent the shortest path.

- String Matching: Naive String Matcher and KMP algorithms are merged together as they contain the same input/output. We keep the "Naive String Matcher" name to represent the string matching task.

- Convex Hull computation: Jarvis March and Graham Scan are combined and represented as "Graham Scan".

Note that some other algorithms might also seem to have the same computation (*e.g.*, MST Prim and MST Kruskal both compute minimum spanning tree), however, they are implemented differently in the CLRS benchmark and we keep them different as well.

## A.2    Leveraging Intermediate Steps of Algorithms

The CLRS benchmark contains intermediate steps of algorithms (*i.e.*, hints) as optional supervision in addition to the supervision of the final output of the algorithm. Namely, for applying an algorithm to a particular graph, two additional supervision signals are provided: (1) number $T$ representing the number of recurrent processing steps to perform (*i.e.*, the processor is called $T$ times), (2) depending on the algorithm, several hints are provided. Each hint is a $T$-dimensional tensor, representing the intermediate output of an algorithm for each intermediate step. For instance, for the Dijkstra algorithm, one hint is the computed distance for each node at each individual step (*i.e.*, $X_{\text{hint}} \in \mathbb{R}^{T \times n}$ where $X_{\text{hint}_{tv}}$ represents the distance of node $v$ to the source node at step $t$). Hints also follow the encode-process-decode framework and have separate encoders and decoders (while sharing the same processor with the inputs and outputs). When leveraging hints, Eq. (1) should be modified to the following formula:

$$H^{(t+1)} = \text{GNN}([\ H^{(t)}, \quad Z_{\text{inp}_v}, \quad Z^{(t)}_{\text{hint}_v}\ ], A), \tag{2}$$

where $Z^{(t)}_{\text{hint}_v}$ is the encoded hints at step $t$ ($1 \leq t \leq T$). The hints for the successor step are also decoded and evaluated in the final loss function along with the final output of the algorithm.

Table 8 contains the full OOD test accuracies of the algorithms when using hints and when disabling them. As shown by the results, enabling hints mostly degrade the final OOD accuracy. While (Veličković et al., 2021) employed a complex model selection on the validation numbers for the with-hint version of processors, running a no-hint version of the simplest form of a processor and no hyperparameter tuning yields superior OOD results.

## A.3    Encode-Process-Decode Framework

Here, we explain how three algorithms are put into a simplified encode-process-decode framework.

**Sorting.** Given a sequence $a_1, a_2, \ldots, a_n$ of scalars, the task is to find the successor of each element when the sequence is sorted. So, the input is again a two-dimensional node feature matrix $X_\mathcal{V} \in \mathbb{R}^{n \times 2}$ where the first dimension denotes the index of each element and the second dimension is the value $a_i$ of that dimension. Here, we do not have an underlying graph and one could consider a fully-connected graph where all nodes are connected to each other. The encoding, processing, and decoding steps are similar to the case of BFS since the final output is again pointer prediction for each node.

**Minimum Spanning Tree (MST) Prim.** Given a weighted graph, the goal is to find the minimum spanning tree of the input graph. A spanning tree is a subset of edges with the lowest sum of weights, which preserve the connectivity of the graph. The input to this task is a two-dimensional node feature matrix $X_\mathcal{V} \in \mathbb{R}^{n \times 2}$ where the first dimension is the node index, and the second dimension indicated the starting node of the MST Prim algorithm. Additionally, the input contains an adjacency matrix $A \in \mathbb{R}^{n \times n}$ and edges features/weights $X_\mathcal{E} \in \mathbb{R}^{n \times n \times 1}$. Again, the procedure is similar to that of BFS, but with the following differences: during the encoding part, we also need an edge encoder (e.g., $W_\text{E} \in \mathbb{R}^{1 \times d_h}$ ) to encode edge features and incorporate them in the processing step. Finally, the decoder needs to predict a binary output for each edge (*i.e.*, whether the edge is in MST or not). So, for each edge $e_{ij}$ with two endpoint nodes $i, j$, respectively, we directly apply a sigmoid function to the output of the decoder $\alpha_{ij}$, and compute the binary cross entropy loss with the ground truth edge label.

## A.4    Alternative Representation for Input Index Representation Shift

The advantage of the Random Scalar Index is that it is a simple extension of the position index into node embeddings. However, it requires a large number of training steps for the neural network to learn the ordering of numbers in the range $[0, 1]$. Ideally, a representation method for ordering of nodes must be invariant under shifting and scaling of the index (*i.e.*, if we shift the index of nodes by some constant factor, the function must not change).

Table 7: Performance of different positional encoding strategies on positions-order aware algorithms for MPNN-FC on L-CLRS dataset.

| Method | Scalar | Random Scalar | Edge Pos. |
|---|---|---|---|
| Dfs | 15.07± 0.92 | 25.29± 1.25 | 28.19± 1.20 |
| Find Maximum Subarray Kadane | 23.81± 2.12 | 48.99± 2.78 | 46.27± 5.29 |
| Dag Shortest Paths | 99.67± 0.12 | 99.72± 0.06 | 99.71± 0.04 |
| Naive String Matcher | 11.51± 4.26 | 7.13± 3.35 | 6.48± 3.43 |
| Matrix Chain Order | 65.19± 3.05 | 91.50± 0.59 | 80.92± 1.71 |
| Topological Sort | 68.77± 2.61 | 70.06± 0.53 | 67.25± 0.90 |
| Bfs | 99.90± 0.08 | 100.00± 0.00 | 99.82± 0.10 |
| Mean | 54.85 | **63.24** | 61.23 |

**Edge Position.** A more natural alternative way of enforcing such an embedding is to encode orderings as edge features (Li et al., 2020). Namely, we create an edge feature from node $v_i$ to node $v_j$ as follows:

$$\text{pos}_{v_i, v_j} = \mathbb{1}\left[i < j\right], \quad \forall i, j \in \{1, 2, \ldots, n\}.$$

This way, the ordering of nodes can be determined by performing a topological sort using two steps of message passing on a fully connected graph. The advantage of using such a method is that nodes do not undergo any distribution shift as the graph grows; hence any subgraph of the larger graph stays in distribution. Moreover, the method is invariant to any shift or scaling of indices $i, j$. On the other hand, using this method requires a processor with global message passing steps and requires an implicit extrapolation of the Topological Sort itself (which is itself challenging, according to the Topological Sort task in the benchmark).

In order to keep the expressiveness of the models and preserve the uniqueness of positions on each node, we append a vector of random features to input features of each node for all representation strategies (Kim et al., 2022). Since this method requires a fully-connected MPNN, we run this method on the MPNN-FC processor and report the results in Table 7. The results show that this method is indeed superior to Scalar Index, although it is outperformed by Random Scalar by a narrow margin.

## A.5 Additional Approaches that Failed

Furthermore, we also tried the following ideas (mostly on the DFS algorithm, and mostly towards making hints work better than no-hints), however, they seem to not work:

- Markovian Assumption on hints: One key factor in failure in OOD generalization is a large number of roll-outs mostly present in sequential algorithms). One scenario where hints might come be useful is to consider a Markovian assumption on steps of the algorithm, meaning that each step of the algorithm is independent of previous steps given its previous state. This then allows for step-wise training and makes training easier and less memory/compute hungry. However, this strategy (on algorithms that satisfy markovian property) did not bring any performance gain over non-markovian assumption.

- Choosing a subset of hints: Each algorithm might contain up to 22 different hints with different types. However, not all of them have a positive effect on a GNN for a better prediction of output and might distract the GNN more than they can help. We tried using only a subset of such hints to see if any boost in output happens or not. For the DFS algorithm, we observed that the fewer hints we use, the higher the accuracy becomes, however, it never surpasses the no-hint version.

- Using hints only as masks: Since an algorithm can contain several hints and not all of them are useful to the GNN, we hypothesized that perhaps a GNN should have fewer restrictions. We used the change in hints as a mask to supervise the GNN for a sparser hidden-representation update. However, again it did not bring any performance gain over the no-hint counterpart and the GNN (on the DFS

Table 8: Full test results of Table 1.

| Method | MPNN-FC | | PGN | | GAT | |
|---|---|---|---|---|---|---|
| Use Hints | Yes | No | Yes | No | Yes | No |
| Articulation Points | 50.91± 2.18 | 50.36± 3.02 | 49.53± 2.09 | 47.85± 2.59 | 37.76± 1.62 | 49.24± 1.30 |
| Activity Selector | 80.66± 3.16 | 91.49± 1.29 | 66.80± 1.62 | 64.59± 1.26 | 73.23± 1.37 | 66.14± 0.63 |
| Bellman Ford | 92.01± 0.28 | 90.14± 1.17 | 92.99± 0.34 | 91.59± 1.99 | 87.91± 1.19 | 78.12± 0.24 |
| Bfs | 99.89± 0.05 | 99.53± 0.20 | 99.63± 0.29 | 99.79± 0.20 | 99.04± 0.21 | 99.14± 0.68 |
| Binary Search | 36.83± 0.26 | 7.06± 6.87 | 76.95± 0.13 | 29.13± 19.12 | 23.50± 3.12 | 3.58± 0.78 |
| Bridges | 72.69± 4.78 | 30.36± 0.73 | 51.42± 7.82 | 32.90± 0.99 | 25.64± 6.60 | 31.52± 0.49 |
| Bubble Sort | 5.27± 0.60 | 50.13± 3.33 | 6.01± 1.95 | 54.56± 4.23 | 9.91± 1.77 | 57.28± 1.55 |
| Dag Shortest Paths | 96.24± 0.56 | 97.23± 0.17 | 96.94± 0.16 | 97.46± 0.35 | 81.14± 1.37 | 80.60± 1.48 |
| Dfs | 6.54± 0.51 | 15.27± 5.29 | 8.71± 0.24 | 22.64± 3.12 | 11.78± 2.04 | 11.13± 1.31 |
| Dijkstra | 91.50± 0.50 | 93.51± 0.46 | 83.45± 1.75 | 93.80± 0.22 | 58.01± 0.79 | 83.27± 0.17 |
| Find Maximum Subarray Kadane | 20.30± 0.49 | 21.65± 0.58 | 65.23± 2.56 | 15.45± 0.31 | 24.43± 0.43 | 14.88± 0.52 |
| Floyd Warshall | 26.74± 1.77 | 39.47± 4.43 | 28.76± 0.51 | 35.19± 1.20 | 16.66± 3.14 | 14.78± 3.85 |
| Graham Scan | 91.04± 0.31 | 96.27± 0.27 | 56.87± 1.61 | 57.05± 0.29 | 77.89± 2.70 | 57.62± 0.33 |
| Heapsort | 10.94± 0.84 | 63.72± 6.41 | 5.27± 0.18 | 55.18± 2.05 | 10.35± 1.83 | 59.55± 1.64 |
| Insertion Sort | 19.81± 2.08 | 38.41± 18.19 | 44.37± 2.43 | 68.77± 2.56 | 29.52± 1.87 | 40.93± 18.49 |
| Jarvis March | 34.86± 12.39 | 95.46± 1.01 | 49.19± 1.07 | 56.64± 0.55 | 51.51± 10.25 | 57.63± 0.33 |
| Kmp Matcher | 2.49± 0.86 | 9.08± 2.39 | 2.00± 0.12 | 2.18± 0.53 | 3.03± 0.36 | 2.28± 0.27 |
| Lcs Length | 53.23± 0.36 | 56.93± 2.87 | 56.82± 0.21 | 60.27± 0.35 | 57.88± 1.02 | 59.70± 0.68 |
| Matrix Chain Order | 79.84± 1.40 | 78.13± 6.56 | 83.91± 0.49 | 82.01± 0.59 | 78.19± 3.31 | 75.13± 5.57 |
| Minimum | 85.34± 0.88 | 99.53± 0.06 | 87.71± 0.52 | 98.01± 1.13 | 84.20± 2.95 | 93.52± 4.88 |
| Mst Kruskal | 70.97± 1.50 | 73.22± 0.39 | 66.96± 1.36 | 71.25± 1.48 | 65.72± 0.99 | 71.57± 2.81 |
| Mst Prim | 69.08± 7.56 | 74.63± 3.19 | 63.33± 0.98 | 73.16± 1.08 | 38.20± 4.34 | 45.46± 4.03 |
| Naive String Matcher | 3.92± 0.30 | 4.43± 0.84 | 2.08± 0.20 | 1.94± 0.27 | 3.01± 1.20 | 1.30± 0.20 |
| Optimal Bst | 62.23± 0.44 | 56.02± 13.13 | 71.01± 1.82 | 30.25± 30.03 | 65.49± 1.75 | 30.42± 2.85 |
| Quickselect | 1.43± 0.69 | 8.50± 3.29 | 3.66± 0.42 | 9.10± 1.07 | 4.36± 0.95 | 8.69± 0.99 |
| Quicksort | 11.30± 0.10 | 58.30± 4.20 | 6.17± 0.15 | 63.87± 3.06 | 7.60± 0.98 | 59.29± 0.92 |
| Segments Intersect | 93.44± 0.10 | 93.94± 0.29 | 77.51± 0.75 | 77.67± 0.92 | 90.41± 0.04 | 76.16± 0.73 |
| Strongly Connected Components | 24.37± 4.88 | 53.65± 3.52 | 20.80± 0.64 | 57.65± 3.95 | 12.70± 3.12 | 39.86± 2.19 |
| Task Scheduling | 84.11± 0.32 | 82.86± 0.32 | 84.89± 0.91 | 81.88± 0.11 | 84.69± 2.09 | 81.93± 0.14 |
| Topological Sort | 52.60± 6.24 | 72.03± 0.74 | 60.45± 2.69 | 78.76± 2.69 | 27.03± 6.92 | 35.30± 19.58 |
| Mean | 51.02±0.60 | **60.04±0.93** | 52.31±0.35 | **57.02±1.23** | 44.69±0.58 | **49.53±0.97** |
| Wins | 8/30 | **22/30** | 10/30 | **20/30** | 14/30 | **16/30** |
| Validation Score Mean | **96.63** | 93.83 | **89.47** | 83.79 | **95.66** | 78.83 |

algorithm) violated the mask quite often (in favor of always updating hidden representations). Even increasing the mask loss to prevent such violations, reduces learning accuracy.

- Discretize/Quantize hidden representations: Due to the long roll-out of many algorithms, small errors might accumulate throughout the processing steps and lead to wrong outputs. We tried quantization techniques such as Straight Through (Bengio et al., 2013), Gumbel-Softmax (Jang et al., 2017), Vector Quantization (van den Oord et al., 2017) on the hidden representation of each processing step. However, training deep quantized networks in an end-to-end fashion hurts the in-distribution training accuracy and even in cases where it did not, OOD accuracy was no better than the continuous counterpart. Even when using hints and trying a Markovian assumption, quantization did not offer any accuracy gain.

Table 9: Validation scores of Table 5.

| Method | MPNN-G | 2WL | Average Hybrid | Sigmoid Hybrid |
|---|---|---|---|---|
| Articulation Points | 100.00± 0.00 | 100.00± 0.00 | 100.00± 0.00 | 100.00± 0.00 |
| Activity Selector | 99.49± 0.18 | 99.49± 0.48 | 99.23± 0.31 | 99.10± 0.18 |
| Bellman Ford | 100.00± 0.00 | 100.00± 0.00 | 100.00± 0.00 | 100.00± 0.00 |
| Bfs | 100.00± 0.00 | 100.00± 0.00 | 100.00± 0.00 | 100.00± 0.00 |
| Binary Search | 99.10± 0.08 | 99.41± 0.11 | 99.45± 0.15 | 99.09± 0.22 |
| Bridges | 99.08± 0.08 | 100.00± 0.00 | 100.00± 0.00 | 100.00± 0.00 |
| Dag Shortest Paths | 100.00± 0.00 | 99.80± 0.28 | 100.00± 0.00 | 100.00± 0.00 |
| Dfs | 99.93± 0.09 | 99.67± 0.18 | 99.22± 0.55 | 99.80± 0.16 |
| Find Maximum Subarray Kadane | 97.51± 0.21 | 97.46± 0.67 | 97.46± 0.18 | 97.48± 0.05 |
| Floyd Warshall | 88.77± 0.90 | 87.37± 0.59 | 88.46± 0.63 | 87.46± 0.42 |
| Graham Scan | 100.00± 0.00 | 100.00± 0.00 | 100.00± 0.00 | 100.00± 0.00 |
| Lcs Length | 75.00± 0.35 | 100.00± 0.00 | 99.98± 0.02 | 100.00± 0.00 |
| Matrix Chain Order | 99.52± 0.03 | 99.33± 0.06 | 99.49± 0.12 | 99.45± 0.10 |
| Minimum | 99.97± 0.02 | 99.97± 0.05 | 99.98± 0.02 | 99.97± 0.05 |
| Mst Kruskal | 92.90± 0.08 | 99.19± 0.33 | 99.77± 0.17 | 99.88± 0.17 |
| Mst Prim | 99.35± 0.18 | 99.35± 0.09 | 99.61± 0.16 | 98.96± 0.18 |
| Naive String Matcher | 80.89± 14.72 | 78.45± 10.80 | 83.15± 4.53 | 92.01± 4.73 |
| Optimal Bst | 96.23± 0.13 | 95.77± 0.17 | 96.30± 0.08 | 95.71± 0.51 |
| Quickselect | 89.19± 5.67 | 99.82± 0.02 | 99.80± 0.07 | 99.67± 0.13 |
| Quicksort | 100.00± 0.00 | 100.00± 0.00 | 100.00± 0.00 | 100.00± 0.00 |
| Segments Intersect | 98.49± 0.10 | 96.42± 0.53 | 97.23± 0.47 | 96.86± 0.67 |
| Strongly Connected Components | 100.00± 0.00 | 99.28± 0.49 | 100.00± 0.00 | 100.00± 0.00 |
| Task Scheduling | 99.50± 0.07 | 99.90± 0.14 | 99.95± 0.07 | 99.85± 0.12 |
| Topological Sort | 100.00± 0.00 | 100.00± 0.00 | 100.00± 0.00 | 99.97± 0.05 |
| Mean | 96.46 | 97.95 | 98.30 | 98.55 |

Table 10: Validation scores of Table 2.

| Method | CLRS-Len-Deg | CLRS-Len | CLRS-Deg |
|---|---|---|---|
| Bellman Ford | 99.81± 0.02 | 99.82± 0.02 | 99.82± 0.02 |
| Bfs | 100.00± 0.00 | 100.00± 0.00 | 100.00± 0.00 |
| Dag Shortest Paths | 99.98± 0.01 | 99.98± 0.01 | 99.98± 0.01 |
| Dfs | 100.00± 0.00 | 100.00± 0.00 | 99.63± 0.06 |
| Floyd Warshall | 87.03± 0.52 | 86.58± 0.40 | 78.40± 0.98 |
| Mst Kruskal | 62.46± 1.09 | 62.22± 0.92 | 63.41± 0.87 |
| Mst Prim | 98.56± 0.07 | 98.43± 0.21 | 97.36± 0.44 |
| Strongly Connected Components | 99.86± 0.05 | 99.83± 0.01 | 99.66± 0.07 |
| Topological Sort | 99.94± 0.03 | 99.93± 0.03 | 99.69± 0.06 |
| Mean | 94.18 | 94.09 | 93.11 |

Table 11: Test scores of proposed processors on 12 representative algorithms on (low-data regime) CLRS dataset. While the 2WL processor shows some level of robustness to the low-data regime, MPNN outperforms 2WL in shortest-path-related tasks since max aggregation inductive bias is baked into the architecture, while the 2WL architecture requires a higher amount of data to learn such inductive biases. However, the hybrid processor still generalizes better compared to both processors.

| Method | MPNN-G | 2WL | Average Hybrid |
|---|---|---|---|
| Articulation Points | 49.78± 0.64 | 48.66± 6.92 | 51.74± 3.83 |
| Bellman Ford | 96.16± 0.88 | 92.55± 1.50 | 91.02± 3.16 |
| Bfs | 100.00± 0.00 | 99.67± 0.17 | 100.00± 0.00 |
| Dag Shortest Paths | 98.45± 0.20 | 88.64± 4.73 | 96.22± 1.61 |
| Dfs | 26.19± 1.20 | 25.29± 9.87 | 27.08± 1.54 |
| Floyd Warshall | 25.92± 2.37 | 21.74± 6.25 | 28.94± 1.23 |
| Graham Scan | 96.81± 0.79 | 73.80± 3.69 | 96.88± 0.35 |
| Lcs Length | 56.60± 1.71 | 86.58± 0.43 | 87.54± 0.39 |
| Matrix Chain Order | 94.81± 0.76 | 83.66± 5.27 | 95.24± 1.29 |
| Mst Prim | 64.34± 6.82 | 53.47± 0.60 | 59.70± 2.69 |
| Quicksort | 48.73± 8.64 | 42.77± 15.35 | 50.99± 28.89 |
| Task Scheduling | 83.10± 0.21 | 82.47± 0.41 | 83.10± 0.21 |
| Mean | 70.07 | 66.61 | 72.37 |

Table 12: Test scores of different hybrid models for combinations of MPNN-G and 2WL on L-CLRS dataset. A hybrid model of two different processors brings accuracy gain over each individual processor.

| Method | Average (MP-MP) | Avergae (2WL-2WL) | Avergae (MP-2WL) | Sigmoid (MP-2WL) |
|---|---|---|---|---|
| Mean | 68.77±0.61 | 69.57±0.64 | 70.92±0.69 | **71.98±1.94** |

Table 13: Full test scores of Table 5 for Scalar Index.

| Method | MPNN-G | 2WL | Average Hybrid | Sigmoid Hybrid |
|---|---|---|---|---|
| Articulation Points | 88.92± 3.39 | 69.59± 5.05 | 78.58± 1.48 | 80.70± 2.11 |
| Activity Selector | 95.81± 0.67 | 90.15± 1.03 | 90.43± 2.65 | 89.37± 2.31 |
| Bellman Ford | 98.83± 0.18 | 98.54± 0.60 | 98.19± 0.14 | 98.03± 0.68 |
| Bfs | 100.00± 0.00 | 100.00± 0.00 | 100.00± 0.00 | 100.00± 0.00 |
| Binary Search | 93.20± 2.07 | 90.30± 1.32 | 86.17± 11.38 | 89.70± 2.80 |
| Bridges | 89.17± 2.13 | 85.22± 11.89 | 98.69± 0.51 | 82.57± 17.44 |
| Dag Shortest Paths | 99.72± 0.06 | 95.91± 1.54 | 99.10± 0.14 | 99.67± 0.06 |
| Dfs | 25.03± 1.62 | 21.11± 4.18 | 19.04± 1.79 | 22.56± 1.08 |
| Find Maximum Subarray Kadane | 26.48± 1.50 | 44.03± 0.53 | 46.01± 2.10 | 43.68± 4.64 |
| Floyd Warshall | 35.39± 2.05 | 30.75± 0.27 | 34.38± 0.98 | 33.59± 1.36 |
| Graham Scan | 98.88± 0.25 | 82.69± 9.57 | 95.26± 1.30 | 92.93± 1.87 |
| Lcs Length | 58.85± 0.80 | 87.06± 0.71 | 86.97± 0.54 | 88.56± 0.32 |
| Matrix Chain Order | 73.65± 9.55 | 76.29± 3.82 | 73.47± 5.28 | 68.70± 3.11 |
| Minimum | 99.56± 0.04 | 99.15± 0.22 | 99.46± 0.11 | 99.58± 0.20 |
| Mst Kruskal | 72.07± 1.03 | 84.77± 2.24 | 90.64± 0.79 | 88.01± 2.31 |
| Mst Prim | 83.84± 1.47 | 75.05± 2.91 | 78.37± 1.04 | 78.03± 2.18 |
| Naive String Matcher | 7.36± 1.64 | 8.48± 1.29 | 9.54± 2.72 | 12.83± 1.30 |
| Optimal Bst | 61.43± 11.78 | 75.94± 0.82 | 74.96± 1.87 | 76.12± 2.06 |
| Quickselect | 3.74± 5.29 | 0.20± 0.21 | 0.23± 0.18 | 29.36± 41.52 |
| Quicksort | 10.06± 3.49 | 17.25± 12.53 | 10.86± 5.96 | 14.45± 7.63 |
| Segments Intersect | 98.16± 0.20 | 96.58± 0.64 | 97.51± 0.45 | 96.37± 0.49 |
| Strongly Connected Components | 75.15± 0.33 | 55.99± 1.79 | 72.15± 1.42 | 73.83± 3.12 |
| Task Scheduling | 83.64± 0.14 | 83.85± 0.55 | 84.21± 0.72 | 83.82± 0.24 |
| Topological Sort | 82.61± 1.50 | 62.09± 8.14 | 77.86± 3.26 | 85.00± 2.53 |
| Mean | 69.23±0.97 | 67.96±0.64 | 70.92±0.73 | **71.98±1.94** |

Table 14: Full test scores of Table 5 for Random Scalar Index. The Weighted Mean is average over all 30 algorithms when considering repeated algorithms with higher weight.

| Method | MPNN-G | 2WL | Average Hybrid | Sigmoid Hybrid |
|---|---|---|---|---|
| Articulation Points | 89.86± 1.79 | 72.46± 2.51 | 79.07± 0.88 | 81.97± 2.99 |
| Activity Selector | 94.02± 1.96 | 89.94± 1.02 | 92.31± 1.40 | 91.07± 1.26 |
| Bellman Ford | 98.84± 0.25 | 98.58± 0.38 | 98.16± 0.15 | 96.74± 2.57 |
| Bfs | 100.00± 0.00 | 100.00± 0.00 | 100.00± 0.00 | 100.00± 0.00 |
| Binary Search | 95.41± 1.35 | 94.94± 0.96 | 94.45± 0.90 | 96.70± 2.15 |
| Bridges | 77.95± 8.57 | 77.96± 9.75 | 97.05± 2.08 | 96.43± 0.82 |
| Dag Shortest Paths | 99.76± 0.07 | 96.60± 1.52 | 99.40± 0.14 | 99.58± 0.20 |
| Dfs | 30.55± 1.85 | 37.50± 2.25 | 26.11± 1.71 | 25.42± 0.78 |
| Find Maximum Subarray Kadane | 49.25± 3.93 | 74.30± 3.57 | 75.70± 1.99 | 69.29± 5.46 |
| Floyd Warshall | 35.04± 1.26 | 30.32± 0.42 | 33.75± 0.90 | 32.66± 1.80 |
| Graham Scan | 98.41± 0.20 | 92.27± 2.40 | 95.80± 0.77 | 93.00± 1.80 |
| Lcs Length | 58.96± 0.53 | 88.40± 1.03 | 87.53± 0.76 | 88.99± 0.60 |
| Matrix Chain Order | 92.34± 0.53 | 91.59± 1.08 | 92.54± 2.17 | 92.12± 1.05 |
| Minimum | 99.59± 0.19 | 99.43± 0.12 | 99.46± 0.16 | 99.50± 0.08 |
| Mst Kruskal | 72.78± 0.50 | 87.56± 2.75 | 90.42± 1.75 | 86.69± 2.59 |
| Mst Prim | 81.79± 1.24 | 76.64± 2.94 | 78.22± 1.97 | 78.52± 1.50 |
| Naive String Matcher | 5.62± 1.71 | 14.08± 5.14 | 16.24± 6.82 | 14.88± 7.65 |
| Optimal Bst | 76.96± 1.19 | 72.49± 2.43 | 77.68± 1.15 | 76.82± 2.83 |
| Quickselect | 3.34± 4.72 | 0.15± 0.08 | 0.63± 0.69 | 0.05± 0.04 |
| Quicksort | 6.41± 1.98 | 7.41± 1.56 | 15.33± 8.84 | 26.24± 19.85 |
| Segments Intersect | 97.74± 0.23 | 96.17± 0.32 | 97.55± 0.78 | 97.34± 0.55 |
| Strongly Connected Components | 75.21± 1.48 | 50.65± 1.37 | 71.47± 1.33 | 72.14± 2.07 |
| Task Scheduling | 84.04± 0.81 | 83.92± 0.58 | 84.12± 0.66 | 83.69± 0.28 |
| Topological Sort | 82.80± 0.96 | 83.18± 1.36 | 80.84± 1.84 | 83.65± 1.26 |
| Mean | 71.11±0.57 | 71.52±0.53 | **74.33±0.49** | 74.31±0.97 |
| Weighted Mean | 64.29 | 64.79 | 68.00 | **68.89** |

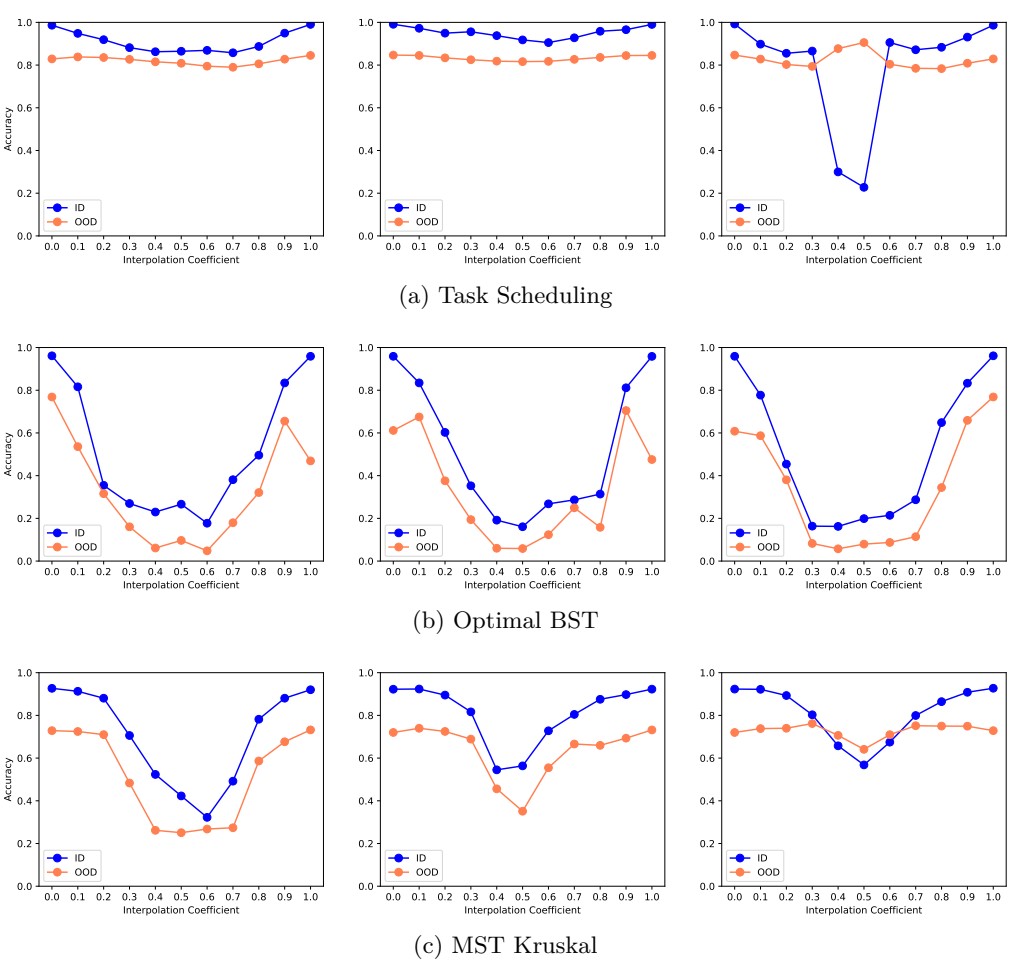

(a) Task Scheduling

(b) Optimal BST

(c) MST Kruskal

Figure 5: Model interpolation charts on three additional tasks. For each task, we train three models and pairwise interpolate their weights and plot ID and OOD accuracies. . Each plot shows the interpolation of one pair for each task. As suggested by the plots, ID and OOD accuracies have very different landscapes and ID accuracy is a suboptimal model selector. In (a), the right plot shows the highest OOD accuracy corresponds with the lowest ID accuracy. In (b), the middle plot achieves its highest OOD accuracy when ID accuracy is not at its peak. In (c), the right plot shows different rates of change for ID and OOD accuracies, OOD accuracy is at its peak when ID is not, and for coefficient 0.5, the OOD accuracy is higher than that of ID.

