# OpenReview forum: "Towards Better Out-of-Distribution Generalization of Neural Algorithmic Reasoning Tasks"
_TMLR — Accepted by TMLR_

### Review · Reviewer_t7tV · 2022-12-07

**Summary Of Contributions:**

The majority of this paper is arguing that OOD generalization in neural algorithmic reasoning tasks requires fundamentally different considerations that more "traditional" OOD problems such as supervised classification or language processing. They point out several cases where existing approaches for improved generalization (understandably/expectedly) fail for algorithmic reasoning, and give convincing evidence for why many evaluation metrics can also be misleading.

The work also proposes some methods for improving OOD generalization of Transformer GNNs on these tasks, with reasonable results.

**Audience:**

Yes

**Claims And Evidence:**

Yes

**Requested Changes:**

Suggested writing changes aside, I don't see any glaring issues to be fixed before publication. I think the main body of the paper is currently quite a bit longer than it needs to be, and that the writers and their intended audience would be much better served with more aggressive editing and pushing more fine-grained details to the appendix.

**Strengths And Weaknesses:**

This paper is reasonably well written, with extensive discussion and thorough reasoning of various design choices made. While it's not totally clear to me how robust the claimed improvements are, I think the authors adequately address the general difficulty of testing this and take pains to evaluate in a variety of settings.

Furthermore, independent of the "improving OOD performance" contribution, I think the general argument made in this paper---that OOD generalization is a fundamentally different task for neural algorithmic reasoning and requires separate consideration/solutions/evaluation---is a useful one that stands on its own. **In fact, I think this constitutes by far the more interesting contribution**, with the proposed methods at the end of the paper feeling a bit like an afterthought and not as essential to the main story of the paper.

I don't have any major complaints, just some general writing suggestions:

* The introduction and earlier discussion is verbose. Lots of parentheses, repeating points several times when once or twice would suffice, etc. I think it would be more effective (and more approachable to casual readers) to state the main points very explicitly *much* sooner, and delay some of the more detailed discussion for a later section so that readers can get the gist of the paper and skip over the more detailed justification if they so choose. **For example, is it absolutely necessary to introduce "hints" so early in the main body and give experiments showing that they're not very helpful, just to say "we won't be using them for the rest of the paper"?** I think this discussion is lengthy and misplaced; I would suggest a brief sentence explaining why they're not being used, with discussion and experiments pushed to the appendix.

* It is disorienting to reference figures in this discussion which are so far removed from when they are first mentioned. I imagine this could be fixed by shortening discussion as described above, but if you *have* to make a certain point so much earlier than the relevant experiments, I would prefer either moving the figure up, or simply referencing general "supporting evidence" in the relevant *section*, and then only referencing the specific figure once you get to that section. It's a bit unpleasant to click on a reference to a figure and then have to scroll all the way back up, and the need to do this generally indicates to me poor planning in the writing structure.

---

> ### Author Response · Authors · 2023-02-01
> **Response to Reviewer t7tV**
>
> Thank you for your thoughtful review of our paper and positive review comments.
>
> > Q1: Stating main points sooner
>
> In the introduction section, we have stated our contributions with specific pointers to each section. We will also add pointers to each subsection to better direct casual readers to particular subsections.
>
> > Q2: More succinct main body and moving hints to the appendix
>
> We appreciate your feedback and agree that the introduction and earlier discussion could be more concise. We will move the detailed description of how hints work to the appendix. Since the original CLRS entirely relies on the hints, we believe that it is necessary to devote a few sentences and a Table in the main text to justify why we deviate from CLRS in this part.
>
> > Q3: Referencing figures near their corresponding text
>
> Thanks for pointing this out. We will remove the first reference to Figure 3 and instead refer to Section 5 for relevant evidence.

---

### Review · Reviewer_2gZT · 2022-12-21

**Summary Of Contributions:**

This paper studies out-of-distribution generalization in the context of so-called "Algorithmic Reasoning Tasks," which are tasks where one uses a modern machine learning method (e.g., deep neural networks) to implement algorithms for known problems such as shortest-path, sorting, etc. The paper makes note of a few differences between OOD generalization in algorithmic reasoning and OOD generalization in computer vision, and proposes a few changes that improve OOD performance across tasks.

**Audience:**

Yes

**Claims And Evidence:**

Yes

**Requested Changes:**

In addition to answering the questions raised above:

- Improve clarity of writing, and fix various typos throughout the paper (e.g., "underperming -> underperforming", "performances -> performance", etc)
- Add error bars for both tables and graphs (Table 1, Table 5, Figure 2)
- Include in-distribution accuracies in Table 1
- More thorough ablation of the improvements, rather than just showing that they are effective sequentially
- In general, improve the presentation of the results: this includes showing both ID and OOD accuracy for the original algorithms considered in Table 1, then for each suggested improvement, showing a Table in the same format as Table 1 that has the accuracies before and after the proposed improvement. As it stands each paper is formatted totally differently and it's hard to tell where the actual improvements are.
- The mode connectivity study could be more convincing---ways to make it more convincing include studying more tasks, using more models, and providing analogous results for vision.

**Strengths And Weaknesses:**

Strengths:
- The paper lays out its contributions very clearly and was easy to follow from this perspective.
- The paper points out a few concrete differences between typical OOD generalization settings and OOD in neural algorithmic reasoning.
- The suggested improvements do seem to generally improve OOD performance.


Weaknesses:
- The exposition could be improved. For example, for those inexperienced with neural algorithmic reasoning (such as myself), Section 2 does little to explain how tasks such as sorting are actually put into graph form.
- The claim that "model selection is hard" is not presented convincingly, since good model selection only requires monotonicity and not linearity in Figure 1---is it true that picking the best ID model performs very poorly on OOD data?
- The solution in 4.1 (increasing dataset size) seems more like changing the problem than an actual solution---it's not really surprising that more data improves performance. The authors seem to acknowledge this and don't highlight it too much in writing, but it would be interesting to know whether the other suggested improvements (like the new processor, increased representativeness, etc.) depend on this increased dataset size, or whether they also improve things in the low-data regime.
- It is often hard to line up the claims in captions/text with the quantitative results in the corresponding figures. For example, the caption of Figure 2 says "While models achieve similar ID accuracy, their interpolations show very different OOD performances." I agree that the variation in the OOD accuracy does seem to be slightly bigger on average, but I think the conclusion that ID is constant while OOD varies is not true. Similarly, the accuracies in 1b and 1d do seem to fall on the line (and so does 1a prior to the x axis saturating at 100%) but this is not discussed in the caption or surrounding text.

---

> ### Author Response · Authors · 2023-02-01
> **Response to Reviewer 2gZT**
>
>
> Thank you for your thoughtful review of our paper and constructive comments.
>
> > Q1: Improve exposition
>
> Thanks for the constructive comment. We will clarify how tasks are put into this framework. Take the sorting algorithm for example. We explain a simplified architecture The input to this task is an array with $n$ elements to be sorted. The output of this task is to predict a pointer from each element to its successor. The input array is represented as pairs of (index, value), both of which are scalars and constitute a node feature matrix $X_\mathcal{V} \in \mathbb{R}^{n \times 2}$. A (fully connected) graph with $n$ nodes and two-dimensional node features is constructed. The encoder is a simple shared linear layer with a weight matrix $W_\text{enc} \in \mathbb{R}^{2 \times d_h}$, where $d_h$ is the latent embedding size. Next, for the processing step, $t$ recurrent steps of GNN are applied to the latent steps to get $H^{(t)}$. Finally, for each pair of nodes $1 \leq i, j \leq n$, a score $\alpha_{i, j}$ determines how likely $i$ points to $j$ and $y_i = \arg\max_{j} \alpha_{i,j}$ is the predicted output pointer for node $i$. To compute $\alpha_i$, we can apply an MLP to $[{h_i}^{(t)}, {h_j}^{(t)}]$ followed by a (row-wise) softmax non-linearity. The loss for this task is a cross-entropy loss between $\alpha_{i, j}$ the true (one-hot row-wise) pointer scores for each node.
> We will add examples with illustrative figures to better clarify this.
>
> > Q2: “Model selection is hard” is not presented convincingly since linearity is not necessary and monotonicity is sufficient.
>
> We do not claim the model selection is hard due to the nonlinearity of ID vs OOD accuracies (although if we had linearity, model selection would have been easier). Our first claim is that when validation accuracy is nearly perfect (which is the case for most tasks as shown in Table 8), there is no good metric for selecting models. This can be seen across different architectures very boldly. For instance, take the Articulation Points task; all architectures achieve 100% validation accuracy in Table 8, while their OOD performance widely differs in Table 11 – from 69% to 88%. This is also true within the same architecture (Figure 1 (a) the bridges task shows a noisy vertical line in the perfect validation regime). While monotonicity holds to some extent when comparing a randomly-initialized network with a fully trained network, what we care about is distinguishing fully trained networks from each other; hence the perfect validation score regime is what matters for model selection.
>
> In our second claim, we examine models with the same architecture and their mode connectivity to better understand this difference. As shown in Figure 2, the accuracy landscapes are completely different. For instance, in Figure 2 (b), the monotonicity is clearly violated (the first half is monotonically decreasing and the second half is monotonically increasing).
>
> We will clarify this in more detail and add additional mode-connectivity figures in our updated draft.
>
> > Q3: increasing dataset size seems more like changing the problem than an actual solution
>
> We agree that this should not be claimed as a solution to out-of-distribution generalization (e.g., comparing two models trained on different dataset sizes is unfair). We, however, believe that this is a (partial) solution to the problem of the unique specification of an algorithm in a data-driven manner. In other words, while it might not help size-generalizing in general, it helps rule out any in-distribution training issue. That is why we create a new dataset to make sure the only remaining challenge for the model is induction (to larger sizes) rather than struggling with a low amount of in-distribution data. For instance, in a low-data regime, careful tuning of hyperparameters for each task plays an important role, while ideally, the focus should be on a universal set of hyperparameters for all tasks, and improving size generalization. Nevertheless, we will add further experiments for our proposed processors on a low-data regime for some of the representative tasks.
>
> > Q4: Figure 2 says "While models achieve similar ID accuracy, their interpolations show very different OOD performances.”. I agree that the variation in the OOD accuracy does seem to be slightly bigger on average, but the conclusion that ID is constant while OOD varies is not true.
>
> In our experiments, we observe that ID accuracy is often nearly perfect (Table 8), however, the OOD results show very different behaviors.

---

> ### Author Response · Authors · 2023-02-01
> **Response to Reviewer 2gZT (cont.)**
>
> > Q5: The accuracies in 1b and 1d do seem to fall on the line (and so does 1a prior to the x-axis saturating at 100%) but this is not discussed in the caption or surrounding text.
>
> Thanks for mentioning this. We should clarify that while some of the trends are close to being a line, the slope is far away from being one (e.g., see [1] Figure 1). We will add this clarification to the caption as well.
>
>
>
> > Q6: Improve clarity, fix typos, add error bars to tables and graphs, include in-distribution accuracies in Table 1
>
> Thanks for your careful read! We will incorporate your feedback in our updated version.
>
>
> > Q7: More thorough ablation of improvements.
>
> Thank you for highlighting this. We will add further ablations for the low-data regime, in addition to the existing ablations we have in our original submission:  (1) For the improved representation, we have Transformer positional encoding as a baseline (Table 4). (2) For our 2WL processor and the Hybrid variant, we have results for both with and without the improved representation results across all different processor types (Table 5, and for individual tasks Tables 11 and 12). (3) Finally, we have experiments when the Hybrid processor is a Hybrid of the same processor type (Table 10).
>
> > Q8: For each suggested improvement, show a Table that has accuracies before and after the proposed improvement.
>
> Thank you for the suggestion. We will add a single table containing numbers before, and after applying each of the improvements.
>
> > Q9: Mode connectivity study could be more convincing by studying more tasks, using more models, and providing analogous results for vision.
>
> Thank you for the feedback. We will add mode connectivity for additional tasks in the appendix in our updated version.
>
> [1] J. P. Miller, R. Taori, A. Raghunathan, S. Sagawa, P. W. Koh, V. Shankar, P. Liang, Y. Carmon, and L. Schmidt. Accuracy on the line: on the strong correlation between out-of-distribution and in-distribution generalization. (ICML 2021) https://arxiv.org/abs/2107.04649

---

### Review · Reviewer_ie3r · 2023-01-23

**Summary Of Contributions:**

This paper studies Out-Of-Distribution (OOD) generalization problem of Neural Algorithmic Reasoning (NAR). The paper argues that OOD generalization for NAR requires special treatment compared with OOD in other domains such as image, as NAR’s taskonomy prevents the use of standard regularizations technique such as data augmentation.

The author points out several contributing factors that makes OOD in NAR challenging, including suboptimal node indexing causing unnecessary distribution shift, dataset generation not covering all modes of the true distribution, model selection under near perfect ID validation scores, and two entangling factors (degree distribution and graph size) in CLRS benchmark.

To alleviate these issues, the paper proposes three major changes to the pipeline, including 1). a set of enlarged CLRS dataset (L-CLRS) that disentangles degree distribution and graph size, 2). randomized node indexing (random scalar) to avoid overfitting, and 3). A new hybrid message-passing method that combines MPNN-G and 2WL processors.

Experimental results in conducted on the proposed variants of CLRS dataset, where the proposed random node indexing and hybrid processor outperforms prior art (MPNN-G).

**Audience:**

Yes

**Claims And Evidence:**

Yes

**Requested Changes:**

[minor] Adding an illustrative example in the Background section to link the graph representation (node/edge/graph) to their respective components of a program.

**Strengths And Weaknesses:**

strength:

- The paper identifies a series of contributing factors that makes the OOD generalization of NAR challenging.
- The paper proposes a set of expanded CLRS datasets to aid the study of the OOD issue.
- The paper proposes a set of improvements to boost the OOD generalization performance. Specifically, the use of randomized node indexing (random scalar) and the proposed hybridization of 2WL and MPNN-G message-passing processors both exhibit visible improvement gains.

weakness:

- The method for enlarging the dataset seems tailored for some specific predefined OOD scenarios, which might harm the coverage of the empirical analysis and evaluation on top of it.
- While the analysis part of the paper points out that model selection is problematic on CLRS due to near-perfect ID validation accuracy, the validation accuracy on L-CRLS also seems to be near-perfect for most cases (Table 9). I could be missing something, so perhaps the author can clarify.

further comments & questions:

1. In the Background section, when discussing CLRS, I feel the clarity could be further improved without much change. Currently, without going through the original CLRS paper, it is not clear to the reader which component of the program a node/edge represents. Perhaps adding an illustrative example would be sufficient.
2. In table 2, why is the graph level accuracy for Floyd Warshall and Mst Kruskal 0 for all dataset variants?

---

> ### Author Response · Authors · 2023-02-01
> **Response to Reviewer ie3r**
>
> Thank you for your thoughtful review of our paper and positive review comments.
>
> > Q1: The method for enlarging the dataset seems tailored for some specific predefined OOD scenarios, which might harm the coverage of the empirical analysis and evaluation on top of it.
>
> It could. For instance, if we add a low-data regime constraint to the training data, then generating many samples can not be counted as a solution. However, the size-generalization OOD scenario is already so challenging by itself (as can be seen from OOD results in Table 5), that adding more constraints to it or trying to improve several scenarios at once is not feasible. Moreover, as shown in Section 3.5, each OOD scenario requires its own setup and is challenging in its own way, and high benchmark numbers in one setting do not transfer to high benchmark numbers in the other setting. As a result, we kept our focus only on size generalization and eliminated other factors that contributed to low OOD accuracy, but were not directly related to the problem of learning to do induction to larger graph sizes (such as low-data regime). We will clarify this in our paper.
>
> > Q2: While the analysis part of the paper points out that model selection is problematic on CLRS due to near-perfect ID validation accuracy, the validation accuracy on L-CRLS also seems to be near-perfect for most cases (Table 9).
>
> That is true. The purpose of introducing L-CLRS was to provide better coverage of ID data and truly evaluate the capabilities of a model on OOD data rather than dealing with different in-distribution training issues. The issue that the validation set is still a suboptimal model selector still exists and finding a better validation strategy is a potential direction for future work, which we will remark in the paper
>
> > Q3: In table 2, why is the graph level accuracy for Floyd Warshall and Mst Kruskal 0 for all dataset variants?
>
> Zero accuracy at the graph level means that no graph in the test set is classified correctly. Note that this metric is much stricter than node-level accuracy. For instance, if a model classifies exactly 50% of nodes/edges of each graph correctly, then it receives 50% node-level accuracy while receiving zero graph-level accuracy. One reason that these two tasks are particularly getting all zeros is that these tasks are edge prediction tasks. So even if one edge (out of $O(n^2)$ possible edges) is classified incorrectly, a graph is considered to be classified incorrectly. While, the rest of the tasks are node prediction tasks and for the graph to be considered classified correctly, we need all nodes ($O(n)$) to be classified correctly. We further need to clarify that while MST Prim and MST Kruskal are both computing MST, their implementations in CLRS benchmark are highly different, and as a result, their benchmark numbers are also different.
>
> > Q4: Add an illustrative example in the Background section to link the graph representation to their respective components of a program.
>
> Thanks for your suggestion. We will add an illustrative example of how the BFS task is incorporated into the encode-process-decode framework in the Background section. We will also add more examples of different tasks in the appendix.

---

### Author Response · Authors · 2023-02-01
**General Response to Reviewers and Action Editors**

We thank all the reviewers for their feedback and their many constructive suggestions. We are encouraged that they found our paper to have a useful analysis with extensive discussion and thorough evaluation (R1), lays contributions clearly (R2), identifies contributing factors to the OOD generalization issue (R3), and that the proposed improvements to be effective (R1,2,3).

We have incorporated reviewers’ feedback in our updated manuscript. A summary of changes to our manuscript is as follows:

* Added an illustrative figure (Figure 1) for a better exposition of the background section, in addition to several examples in the appendix (A3).
* Added further clarification to “Why model selection is hard” in Section 3.4 accompanied by additional mode connectivity charts in the appendix.
* Moved the description of “how hints work” into the appendix.
* Added error bars to average OOD accuracy results in the appendix.
* Added ablations of our processors for the low-data regime (Table 10).
* Added minor clarifications throughout the text, which were raised by the reviewers.

Please let us know if any part needs more clarification.

Best,
Authors

---

### Decision · Action_Editors · 2023-02-26

**Recommendation:** Accept with minor revision

**Comment:**

See Claims and Evidence above.

**Audience:**

Yes, some individuals in TMLR's audience be interested in knowing the findings of this paper

**Claims And Evidence:**

This paper distinguishes out-of-distribution generalization in neural algorithmic reasoning tasks with standard out-of-distributions problems in ML, and proposes techniques to improve the out-of-distribution generalization ability. The framework follows the standard setup of using GNN for this problem. All reviewers think the paper is presented clearly, and the contribution is enough for publication. They also raise some issues in terms of presentation, clearness, and clarifications for some specific techniques. After rebuttal, all reviewers are satisfied with the paper and recommend acceptance.

I am on part with the reviewers, but also share the concern that the paper should formulate neural algorithmic reasoning clearly at the very beginning of the paper. The current form skips this and make the paper difficult to be accessed to people out of the research domain. Although the authors tried to address the problem by adding most of the clarifications in the appendix, I think the setup should be moved to and revised at the beginning the paper (probably after Introduction). I request the authors to make this change before acceptance of the paper.